# Macrophage secretion of miR-106b-5p causes renin-dependent hypertension

J. Oh[1,11], S. J. Matkovich[1,11], A. E. Riek [1], S. M. Bindom[2], J. S. Shao[1], R. D. Head[3], R. A. Barve [3], M. S. Sands[1,3], G. Carmeliet [4], P. Osei-Owusu[5], R. H. Knutsen[5], H. Zhang[5], K. J. Blumer[5], C. G. Nichols [5], R. P. Mecham [5], Á Baldán [6], B. A. Benitez [7], M. L. Sequeira-Lopez[8,9], R. A. Gomez[8,9] & C. Bernal-Mizrachi [1,5,10]✉

Myeloid cells are known mediators of hypertension, but their role in initiating renin-induced hypertension has not been studied. Vitamin D deficiency causes pro-inflammatory macrophage infiltration in metabolic tissues and is linked to renin-mediated hypertension. We tested the hypothesis that impaired vitamin D signaling in macrophages causes hypertension using conditional knockout of the myeloid vitamin D receptor in mice (KODMAC). These mice develop renin-dependent hypertension due to macrophage infiltration of the vasculature and direct activation of renal juxtaglomerular (JG) cell renin production. Induction of endoplasmic reticulum stress in knockout macrophages increases miR-106b-5p secretion, which stimulates JG cell renin production via repression of transcription factors E2f1 and Pde3b. Moreover, in wild-type recipient mice of KODMAC/miR106b$^{-/-}$ bone marrow, knockout of miR-106b-5p prevents the hypertension and JG cell renin production induced by KODMAC macrophages, suggesting myeloid-specific, miR-106b-5p-dependent effects. These findings confirm macrophage miR-106b-5p secretion from impaired vitamin D receptor signaling causes inflammation-induced hypertension.

[1] Department of Medicine, Washington University School of Medicine, St. Louis, MO, USA. [2] Department of Pediatrics, Washington University School of Medicine, St. Louis, MO, USA. [3] Department of Genetics, Washington University School of Medicine, St. Louis, MO, USA. [4] Laboratory of Clinical and Experimental Endocrinology, Department of Chronic Diseases, Metabolism, and Aging, KU Leuven, Leuven, Belgium. [5] Department of Cell Biology and Physiology, Washington University School of Medicine, St. Louis, MO, USA. [6] Edward A. Doisy Department of Biochemistry and Molecular Biology, Saint Louis University, St. Louis, MO, USA. [7] Department of Psychiatry, Washington University School of Medicine, St. Louis, MO, USA. [8] Department of Pediatrics, University of Virginia School of Medicine, Charlottesville, VA, USA. [9] Department of Biology, University of Virginia Graduate School of Arts and Sciences, Charlottesville, VA, USA. [10] Department of Medicine, VA Medical Center, St. Louis, MO, USA. [11]These authors contributed equally: J. Oh, S.J. Matkovich. ✉email: cbernal@wustl.edu

Hypertension is the most common cardiovascular risk factor and a major contributor to cardiovascular morbidity and mortality[1]. As the cellular and molecular mechanisms of hypertension and inflammation are more clearly defined, it is apparent that these processes are interrelated. Overactivation of the renin–angiotensin system (RAS) creates a cycle of pro-inflammatory macrophage accumulation and stimulation of oxidative stress in the vasculature, resulting in nitric oxide scavenging, low renal blood flow, and subsequently increased renin secretion from the juxtaglomerular apparatus[2]. Conversely, the absence of the monocyte lineage prevents angiotensin II-induced hypertension in mice[3]. Transplantation of wild-type (WT) bone marrow macrophages restores hypertensive responses to angiotensin II in these mice, suggesting that innate immune cells are necessary mediators in the development of hypertension[4]. However, immune cells have never been shown to directly initiate renin-induced hypertension. Observational studies link vitamin D deficiency to the development of hypertension[5]. Mendelian randomization analysis suggests a causal association between increased 25-hydroxyvitamin D [25(OH)D] concentrations and reduced blood pressure and hypertension risk[6]. However, randomized controlled trials have yielded mixed results, likely due to small study size, variable treatment regimens, and inclusion of participants with normal vitamin D levels or antihypertensive medications[7]. In mice, germline vitamin D receptor (VDR) knockouts exhibit hypertension and cardiac hypertrophy due to upregulation of renin promoter activity and RAS activation[8]. The VDR is expressed in myeloid cells and B and T lymphocytes[9]. VDR activation prevents IFNγ-, LPS-, and CD40L-induced monocyte production of pro-inflammatory cytokines known to induce vascular inflammation[10–12]. Vdr deletion promotes a pro-inflammatory macrophage phenotype, with increased adhesion and migration into the vasculature causing atherosclerosis progression[13,14]. In this study, we show that vitamin D-mediated ER stress and subsequent miR-106b-5p secretion from macrophages is sufficient to cause hypertension.

## Results

**Myeloid-specific *Vdr* deletion induces hypertension.** To begin to determine the role of macrophages in the hypertensive phenotype resulting from vitamin D deficiency, we generated myeloid cells lacking VDR (KODMAC) by crossing $Vdr^{fl/fl}$ mice with lysosome-M-promoter-driven Cre mice (*Lyz2Cre*) in the $Ldlr^{-/-}$ background (a model of diet-induced metabolic syndrome), and compared them to $Vdr^{fl/fl}Ldlr^{-/-}$ littermates (control). At 8 weeks, chow-fed KODMAC mice showed increased mean arterial blood pressure by 24-h telemetry in both genders (Fig. 1a), with significant differences in both systolic and diastolic blood pressure (SBP and DBP) that persisted for at least 1 year (Supplementary Fig. 1a, b). KODMAC mice in the $ApoE^{-/-}$ or C57BL6 backgrounds showed similar results (Supplementary Fig. 1c, d). KODMAC mice also had 30% higher plasma renin and lower urinary sodium (Fig. 1b and Supplementary Fig. 1e), indicating increased RAS activation. Assessment of the distal thoracic aorta of 12-week-old, chow-fed KODMAC mice revealed increased vascular macrophage infiltration (Fig. 1c and Supplementary Fig. 1f, g), as well as increased reactive oxygen species formation capable of scavenging nitric oxide in the vessel wall, shown by increased dihydroethidium staining (Fig. 1d), higher NADPH-mediated detection of ROS (Fig. 1e), and lack of BP response to the NO synthase inhibitor L-NAME (Fig. 1f). These events resulted in decreased renal perfusion (Fig. 1g). Lethally irradiated KODMAC mice transplanted with bone marrow from other KODMAC (KOD→KOD) or control mice (Con→KOD) demonstrated that restoration of myeloid VDR could prevent

hypertension, with 20–28 mmHg lower BP (Fig. 1h) and a 36% reduction in plasma renin (Fig. 1i) after 8 weeks in Con→KOD mice compared to KOD→KOD mice. Conversely, control mice transplanted with bone marrow from KODMAC mice KOD→Con had an ~18 mmHg increase in BP compared to (Con→Con) controls (Fig. 1h), with threefold-increased plasma renin (Fig. 1i). These data demonstrate that loss of myeloid VDR signaling causes renin-mediated hypertension.

**Macrophage miR-106b-5p secretion induces renin production.** Previous studies in spontaneous models of hypertension show renal interstitial infiltration of macrophages and lymphocytes prior to the development of hypertension, suggesting that immune cells may be responsible for hypertension[15]. Serial cryosections of KODMAC compared to control littermate kidneys stained with macrophage-specific antibodies (MOMA) (Fig. 2a) that were confirmed by flow cytometry (Supplementary Fig. 2a, b) showed increased macrophages that were adjacent to juxtaglomerular (JG) cells (Fig. 2b). To investigate whether JG-localized macrophages could increase renin production directly, we cocultured KODMAC macrophages with engineered JG cells expressing YFP under control of the renin promoter (Ren1$^c$-YFP)[16,17]. Both KODMAC macrophages and KODMAC media alone were capable of inducing renin production in JG cells as shown by immunofluorescence (Fig. 2c and Supplementary Fig. 2c), as well as renin mRNA expression and secretion compared to control (Supplementary Fig. 2d–g), suggesting that a factor secreted from KODMAC macrophages activates JG cells. Previous studies indicate that locally produced TNFα and IL-1β may play an important paracrine role in the regulation of the RAS[18]. While media from unstimulated KODMAC macrophages had triple the TNFα levels of control media (Fig. 2d), adding anti-TNFα antibody to KODMAC media during coculture with JG cells did not significantly reduce renin production, suggesting an alternate mechanism of RAS stimulation (Fig. 2e).

Recently, the discovery of circulating extracellular microRNAs (miRNAs) has suggested a role for miRNAs as intercellular signaling mediators[19]. Unbiased miRNA expression analyses using media from KODMAC or control peritoneal macrophages identified 361 differentially expressed miRNAs with $P < 0.05$ (Supplementary Fig. 3a, Supplementary Data 1, NCBI GEO repository number GSE 155511). We selected 12 upregulated miRNAs for secondary validation based on prior literature reports of involvement and/or association with diabetes, hypertension, and endothelial function. We then isolated exosomes, verified their purity, and confirmed by qPCR that in nine of these miRNAs, the more abundant product was increased in those from KODMAC media (Fig. 2f and Supplementary Fig. 3b, c). Of these nine miRNAs, five were also increased in media of macrophages from dietary vitamin D-deficient mice compared to vitamin D-sufficient mice (Fig. 2g), with miR-106b-5p being the most differentially secreted. To clarify whether any of these macrophage miRNAs were increasing JG cell renin production and release, we transfected Ren1$^c$-YFP JG cells with mimics of the miRNAs of interest (106b-5p, 340-5p, 193a-3p, 99b-5p, and 325-3p) or miRNA control. We found that all transfected JG cells had high mRNA abundance (Supplementary Fig. 4a), but miR-106b-5p-mimic-transfected cells had fourfold higher renin production compared to cells transfected with nontargeting control, and had higher renin production and secretion compared to all other mimic-transfected cells (Fig. 2h and Supplementary Fig. 4b). Conversely, transfection of Ren1$^c$-YFP JG cells with miR-106b-5p antagomir suppressed renin production induced by KODMAC peritoneal macrophage media, confirming the regulatory role of miR-106b-5p in JG cell renin production (Fig. 2i). Ren1$^c$-YFP JG

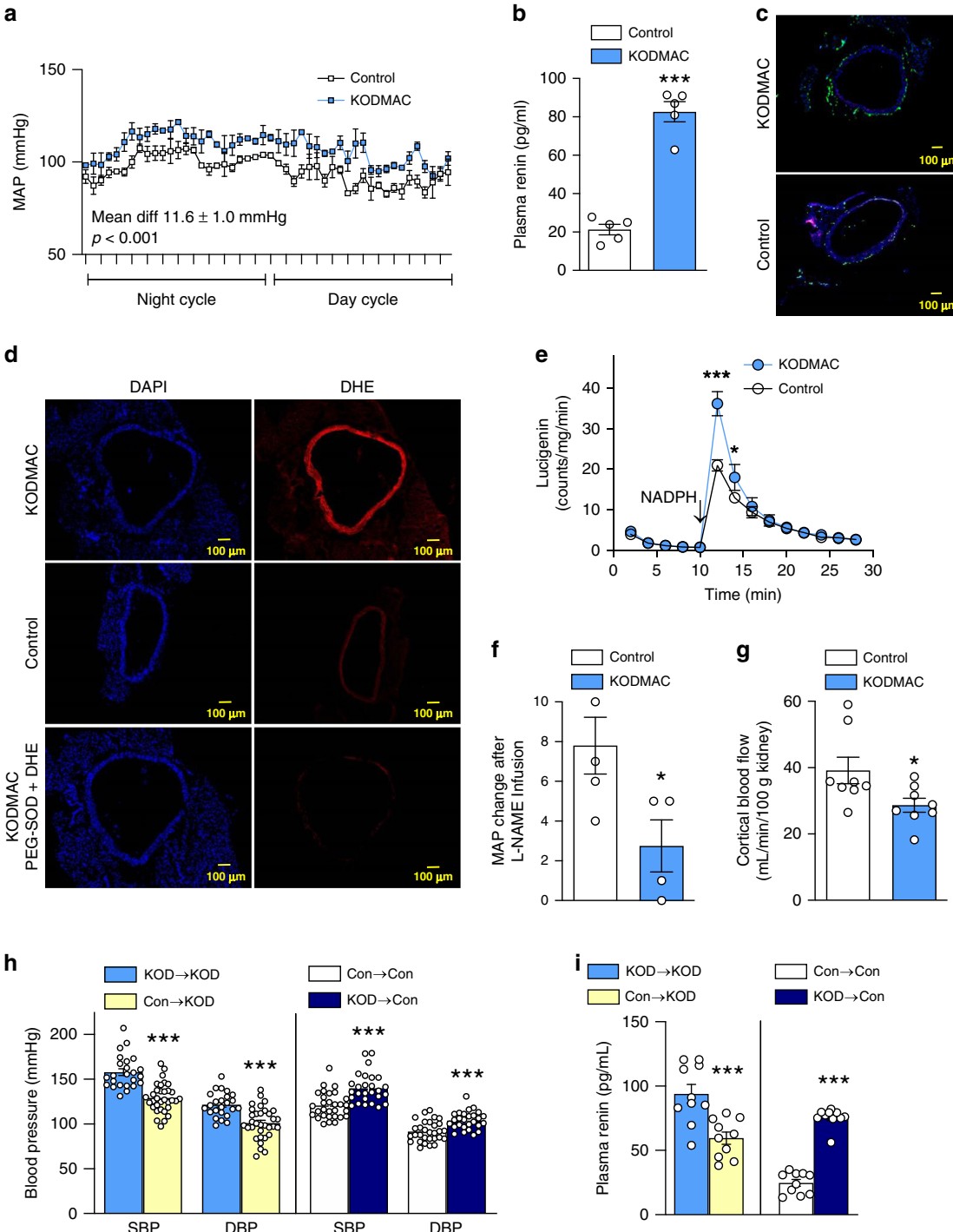

**Fig. 1 Myeloid *Vdr* deletion induces hypertension by increasing vascular ROS.** In 8-week-old KODMAC (light-blue bar) or control mice (white bar): **a** invasive mean arterial BP ($n = 3$/group), **b** plasma renin ($n = 5$/group), distal thoracic aortic assessment of (**c**) MOMA staining (green, DAPI = blue) (representative of four/group, scale bars: 100 μm), **d** DHE staining (red, DAPI = blue) in the presence or absence of PEG-superoxide dismutase (representative of four/group, scale bars: 100 μm), **e** NADPH-stimulated lucigenin-enhanced chemiluminescence ($n = 4$/group), **f** increase in mean arterial blood pressure (MAP) with L-NAME ($n = 5$ Con/4 KOD), and **g** renal cortical blood flow ($n = 8$/group). Irradiated KODMAC mice transplanted with BM from KODMAC (KOD→KOD) (light-blue bar) or control (Con→KOD) (yellow bar) and irradiated control mice transplanted with BM from KODMAC (KOD→Con) (dark-blue bar) or control (Con→Con) (white bar). **h** Noninvasive BP ($n = 24$–33/group) and **i** plasma renin ($n = 5$/group). Data presented as mean ± SEM from (**a**) mixed model analysis, (**b**–**g**) Student's two-tailed unpaired *t* test with *$P < 0.05$, ***$P < 0.001$ vs. control, and (**h**, **i**) Student's two-tailed unpaired *t* test with ***$P < 0.001$ vs. matched donor/recipient control transplant.

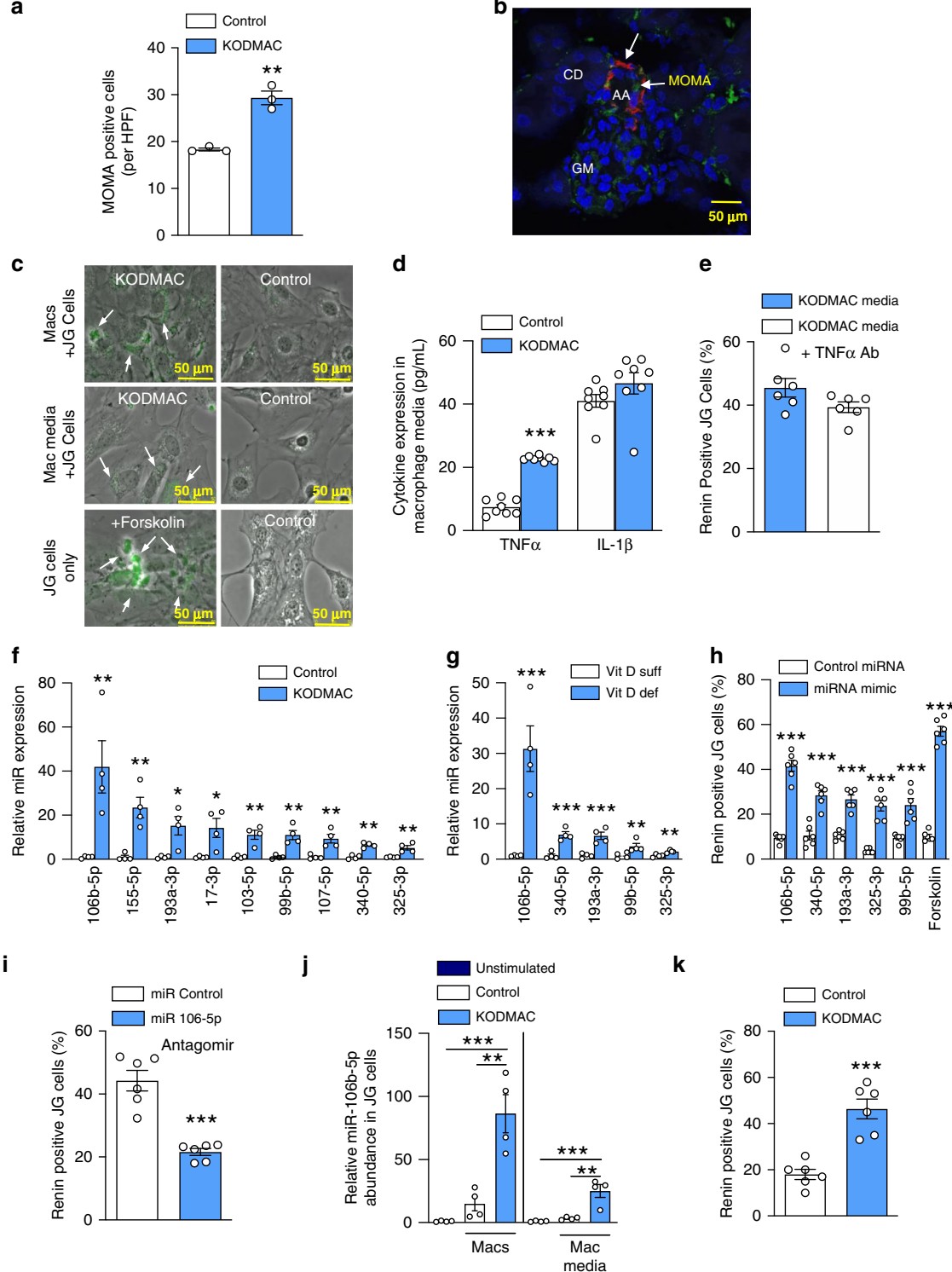

cells expressed little miR-106b-5p at the baseline, but after exposure to KODMAC peritoneal macrophages or their media for 72 h, exhibited a fivefold increase in miR-106b-5p abundance compared to control exposure, suggesting that JG cells take up secreted macrophage miR-106b-5p (Fig. 2j). To confirm that macrophage-secreted miR-106b-5p was the source of increased miR-106b-5p in JG cells, we proactively silenced endogenous miR-106b in Ren1c-YFP JG cells by transfection with siRNA against pre-miR-106b, then cocultured these cells with KODMAC or control macrophage media. KODMAC media exposure

resulted in significantly increased mature miR-106b-5p abundance and renin production in Ren1c-YFP JG cells compared to cells exposed to control media without induction in pre-miR-106b-5p expression (Supplementary Fig. 4c and Fig. 2k). Similarly, KODMAC media increased mature miR-106b-5p abundance in Ren1c-YFP JG cells transfected with siRNA control without changes in pre-miR-106b abundance compared to cells exposed to control media. These data imply that macrophages are the source of miR-106b-5p. Furthermore, media from peritoneal macrophages of dietary vitamin D-deficient mice increased JG

**Fig. 2 KODMAC macrophage miR-106b-5p secretion induces JG cell renin production. a** MOMA-positive cells per high-power field (HPF) in the kidney of KODMAC (light-blue bar) or control mice (white bar) ($n = 3$/group). **b** Confocal microscopy of JG apparatus in KODMAC mice with staining for renin (red), macrophages (green, MOMA-2), and merge (yellow) showing an afferent artery (AA), collecting duct (CD), and glomerulus (GM) (scale bars: 50 μm) (representative of $n = 5$). **c** Renin-positive (YFP-labeled Ren[1c]) JG cells (renin expression = green) cocultured with KODMAC or control macrophages or their media (forskolin = positive control) (representative of 6/group, scale bars: 50 μm). **d** Peritoneal macrophage media cytokine secretion from KODMAC or control mice ($n = 8$/group). **e** Renin-positive JG cells after culture with KODMAC macrophage media with or without anti-TNFα antibody ($n = 6$/group). Relative miRNA expression from (**f**) KODMAC or control peritoneal macrophage media exosomes ($n = 4$/group) and **g** vitamin D-deficient or -sufficient peritoneal macrophage media exosomes ($n = 4$/group). **h** Renin-positive JG cells after transfection with miRNA mimics ($n = 6$/group). **i** Renin-positive JG cells after transfection with miR-106b-5p antagomir or miR control and exposed to KODMAC macrophage media ($n = 6$/group). **j** miR-106b-5p abundance in JG cells before and after coculture with KODMAC or control peritoneal macrophages or their media ($n = 4$/group). **k** Renin positivity in JG cells transfected with pre-miR-106b siRNA, then cocultured with KODMAC or control macrophage media ($n = 6$/group). Data expressed as mean ± SEM from (**a–h**, **i**, **k**) Student's two-tailed unpaired $t$ test with **$P < 0.01$, ***$P < 0.001$ vs. control or vitamin D-sufficient and **j** one-way ANOVA with Tukey's post hoc test with **$P < 0.01$, ***$P < 0.001$.

cell miR-106b-5p abundance and renin production compared to that of vitamin D-sufficient mice (Supplementary Fig. 4d, e), further demonstrating in a more clinically relevant model a significant role for vitamin D-deficient macrophage-derived miR-106b-5p in the stimulation of JG cell renin production.

**Macrophage miR-106b-5p knockout prevents hypertension.** To investigate in vivo the role of macrophage miR-106b-5p secretion in the high blood pressure induced by vitamin D deficiency, miR-106b[−/−] and wild-type littermate mice were fed a vitamin D-deficient or -sufficient chow diet for 10 weeks. As expected, wild-type mice fed a vitamin D-deficient diet had ~10 mmHg higher SBP and DBP compared to wild-type vitamin D-sufficient mice. Importantly, miR-106b knockout prevented the elevation in blood pressure and plasma renin induced by vitamin D deficiency (Fig. 3a and Supplementary Fig. 5a). Media from peritoneal macrophages from vitamin D-deficient miR-106b[−/−] mice as well as from vitamin D-sufficient miR-106b[+/+] and wild-type littermate mice demonstrated significantly lower activation of JG cell renin production when compared to macrophage media from wild-type, vitamin D-deficient littermates (Fig. 3b). Bone marrow (BM) transplant from miR-106b[−/−] mice into hypertensive vitamin D-deficient wild-type mice decreased SBP and DBP and plasma renin, compared to BM transplant from wild-type mice (Fig. 3c, d). To confirm the specificity of miR-106b from KOD-MAC macrophages as the mechanism inducing high BP and JG cell renin production, we generated double-knockout (KOD-MAC/miR-106b[−/−]) mice for comparison to KODMAC/miR-106[+/+] and control Vdr[+/+]/miR-106[+/+] mice and transplanted their BM into WT mice. We found that the absence of miR-106b prevented the SBP and DBP elevation, increased plasma renin, and stimulation of Ren1[c]-YFP JG renin production by KODMAC macrophages, suggesting that KODMAC macrophage secretion of miR-106b-5p increases RAS-dependent hypertension in vivo (Fig. 3e–g).

**Macrophage ER stress causes miR-106b-5p secretion.** Previous studies indicated that vitamin D deficiency or lack of Vdr in macrophages increases both PERK and IRE phosphorylation, as well as Chop protein expression by reducing SERCA function[13,20–22]. Converging evidence indicates that ER stress signaling modulates miRNA expression and accelerates exosome formation and release[23,24]. Therefore, we hypothesized that secretion of miR-106-5p by KODMAC macrophages would be regulated by ER stress. Treatment of vitamin D-deficient macrophages with phenylbutyric acid (PBA, a chemical chaperone known to reduce ER stress) reduced miR-106b-5p secretion into the media by more than 65% and decreased media stimulation of JG cell renin production and secretion (Fig. 3h, i and

Supplementary Fig. 5b). Conversely, induction of macrophage ER stress by thapsigargin in vitamin D-sufficient control macrophages increased miR-106b-5p secretion by 60%, and media from treated cells increased JG cell renin production and secretion (Fig. 3j, k and Supplementary Fig. 5c). Since PBA is a nonspecific inhibitor of ER stress, we obtained mice with knockout of CHOP (Ddit3) and fed them vitamin D-deficient or -sufficient chow diet for 10 weeks. Vitamin D deficiency did not increase SBP or DBP or plasma renin in Ddit3[−/−] mice (Fig. 3l and Supplementary Fig. 5d). Moreover, media from vitamin D-deficient peritoneal macrophages from Ddit3[−/−] mice as well as from vitamin D-sufficient Ddit3[+/+] and Ddit3[−/−] mice showed reduced stimulation of JG cell renin production and 60% less miR-106b-5p secretion compared to vitamin D-deficient Ddit3[+/+] macrophages (Fig. 3m and Supplementary Fig. 5e). BM transplant from Ddit3[−/−] mice into hypertensive vitamin D-deficient Ddit3[+/+] mice decreased SBP, DBP, and plasma renin compared to BM transplant from Ddit3[+/+] mice (Fig. 3n, o), confirming that activation of the ER stress PERK/CHOP pathway by inactivation of macrophage VDR signaling increases miR-106b-5p secretion and renin-dependent hypertension.

**miR-106b-5p activates PPAR and CREB pathways to induce renin.** Renin transcription is controlled by multiple factors capable of binding to either or both an upstream enhancer and the proximal promoter region. Among these, peroxisome proliferator-activated receptor (PPAR) and cAMP-response element-binding protein (CREB) family factors have been previously demonstrated to enhance renin transcription[25]. Global mRNAs were assessed by RNA sequencing analysis for response to miR-106b-5p transfection in JG cells, and renin activators and effectors were confirmed by qPCR, including upregulation of peroxisome proliferator-activated receptor gamma coactivator 1-alpha (Ppargc1a) expression. In addition, protein levels of cAMP-response element-binding protein (CREB) and its activated form, phospho-CREB, were also upregulated, despite variation in CREB family mRNA expression (Fig. 4a and Supplementary Fig. 6). To search for links between the upregulation of these factors and mRNAs likely to be directly targeted for suppression by miR-106b-5p, we performed RISC sequencing from miR-106b-5p-transfected JG cells and identified 1504 mRNAs for which the value of the Ago2:polyA-RNA ratio increased by at least 2-fold upon miR-106-5p transfection, indicating suppression of gene expression (NCBI GEO repository number GSE11770). Ingenuity pathways analysis was used to identify mRNAs from this set of 1504, which were also present in an interaction network focused on Ppargc1a. Of the 18 RISC-enhanced mRNAs present in the Ppargc1a interaction network, E2f1[26] and Pde3b[27] were deemed the most likely to result in Ppargc1a upregulation upon their own suppression. In order to identify mRNAs that would lead to

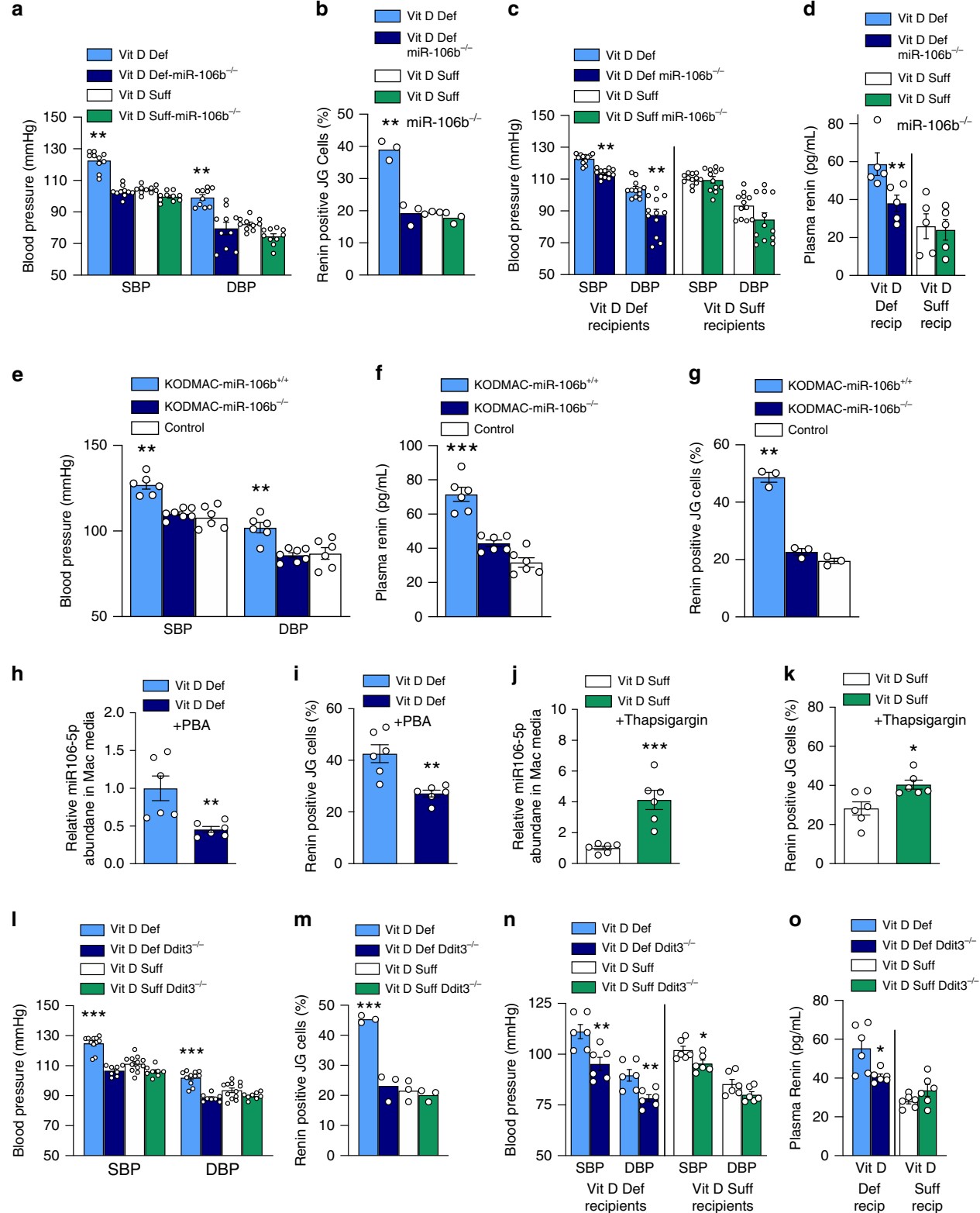

enhanced CREB activity when suppressed, we repeated the analysis with Ingenuity interaction networks focused on adenylate cyclases. miR-106b-5p increased RISC association, implying suppression of gene expression, of *Pde3b* and *Pde10a* (Fig. 4b), which hydrolyze cAMP and thus inhibit CREB activity[27,28]. Furthermore, *E2f1* and *Pde3b*, but not *Pde10a*, have miR-106-5p family-binding sites in both the human and mouse gene 3′UTRs (Supplementary Fig. 7). Ren1^c-YFP JG cells exposed to

KODMAC media had decreased E2F1 and PDE3B protein expression, increased mRNA expression of *Ppargc1a* and *Pparg*, and increased the levels of cAMP, CREB, and phospho-CREB, suggesting that media miR-106b-5p is sufficient to repress the endogenous levels of E2F1 and PDE3B, increasing PPARγ/CREB stimulation of JG cell renin production (Fig. 4c–e). To validate that *E2f1* and *Pde3b* downregulation are critical to JG cell renin production, we knocked down *E2f1* and *Pde3b* in JG cells to

**Fig. 3 Macrophage ER stress regulates miR-106-5p secretion and BP.** In dietary vitamin D-deficient or -sufficient miR-106b$^{-/-}$ or littermates: **a** noninvasive BP ($n = 10$/group) and **b** renin-positive (YFP-labeled Ren1$^{c}$) JG cells ($n = 3$/group) after culture with macrophage media. In vitamin D-deficient (light- and dark-blue bars) or -sufficient control mice (white and green bars) transplanted with BM from miR-106b$^{-/-}$ or control mice on matched diet: **c** noninvasive BP ($n = 12$ recip/group) and **d** plasma renin ($n = 5$ recip/group). In wild-type recipients transplanted with BM from KODMAC/miR-106b$^{-/-}$ (light-blue bar), KODMAC/miR-106$^{+/+}$ (dark-blue bar), or control Vdr$^{+/+}$/miR-106$^{+/+}$ mice (white bar): **e** noninvasive BP ($n = 6$/group), **f** plasma renin ($n = 6$/group), and **g** renin-positive JG cells after culture with macrophage media ($n = 3$/group). In JG cells cocultured with media from vitamin D-deficient peritoneal macrophages treated with (dark-blue bar) or without phenylbutyric acid (PBA) (light-blue bar): **h** miR-106b-5p abundance and **i** renin-positive cells ($n = 6$/group). In JG cells cocultured with media from vitamin D-sufficient peritoneal macrophages treated with (green bar) or without thapsigargin (white bar): **j** miR-106b-5p abundance and **k** renin-positive cells ($n = 6$/group). In Ddit3$^{+/+}$ or Ddit3$^{-/-}$ mice fed vitamin D-deficient and -sufficient diets: **l** noninvasive BP ($n = 8$–11/group), **m** renin-positive JG cells after coculture with their peritoneal macrophage media ($n = 3$/group). In vitamin D-deficient (light- and dark-blue bars) or -sufficient control mice (white and green bars) transplanted with BM from Ddit3$^{-/-}$ or control mice on matched diet: **n** noninvasive BP ($n = 6$ recip/group) and **o** plasma renin activity ($n = 6$ recip/group). Data expressed as mean ± SEM from (**a**, **b**, **e**–**g**, **l**, **m**) one-way ANOVA with Tukey's post hoc test with ***$P < 0.001$ vs. all, (**c**, **d**, **n**, **o**) Student's two-tailed unpaired $t$ test with *$P < 0.05$, **$P < 0.01$, ***$P < 0.001$ vs. matched donor/recipient diet, and (**h**–**k**) Student's two-tailed unpaired $t$ test with *$P < 0.05$, **$P < 0.01$, ***$P < 0.001$ vs. vitamin D-deficient or -sufficient macrophage control media.

attempt to mimic, at least in part, the likely effect of miR-106b-5p on increased renin production. Transfection of Ren1$^{c}$-YFP JG cells with siRNAs against either E2f1 or Pde3b reduced mRNA levels (Supplementary Fig. 8) and increased renin production by approximately fourfold and secretion by more than twofold (Fig. 4f, g) compared to siRNA control-transfected cells, similarly to coculture with KODMAC macrophage media. In summary, impaired vitamin D receptor signaling in macrophages results in ER stress that increases miR-106b-5p secretion. This miRNA is taken up by JG cells to activate renin production, identifying a mechanism of inflammation-induced hypertension (Fig. 4h).

## Discussion

We postulate that innate cellular immunity orchestrates the interaction between metabolic tissues and environmental factors involved in the development of hypertension. Vitamin D deficiency promotes activation of innate immunity, increasing cytokine production and accelerating vascular infiltration of immune cells[14,29]. However, it is unclear whether the pro-inflammatory changes induced by vitamin D deficiency can accelerate hypertension. Early studies suggested that direct VDR activation in JG cells repressed renin promoter activation[8]. In this study, we found that myeloid-specific VDR deficiency is sufficient to induce high blood pressure in mice with adequate vitamin D status. In three different mouse models, we found that deletion of myeloid Vdr promotes vascular and renal macrophage infiltration and increases RAS-dependent hypertension. Induction of endoplasmic reticulum stress in KODMAC macrophages increases exosome secretion of miR-106b-5p, which induces JG cell renin production via repression of transcription factors E2f1 and Pde3b. Recipients of BM from KODMAC mice experience increased BP via activation of renal renin production, effects that are prevented by the knockout of miR-106b-5p in the transplanted cells, together confirming a microRNA-dependent macrophage-specific communication with JG cells causing renin-mediated hypertension.

The role of lymphocytes in the development of hypertension is well-established; nonetheless, activation or expansion of lymphocytes requires activation of the innate immune system[30]. Previous studies have shown roles for macrophages in mediating Ang II-induced or DOCA-salt-induced hypertension[15,31], but it is unknown whether their effects are sufficient to initiate hypertension. Maternal exposure to a low-protein diet induces infiltration by immune cells and ROS production in the kidneys of Sprague–Dawley rat offspring before hypertension develops, and these events are prevented by early exposure to antioxidants and immunosuppressive medication[32]. Moreover, a missense mutation of SH2B adaptor protein 3 in immune cells prevents

infiltration of leukocytes into the kidneys and high-salt-diet-induced hypertension in the Dahl salt-sensitive rat, further indicating a potential role of immune cells early in the pathogenesis of hypertension[33]. In this study, we discovered that pro-inflammatory macrophages are capable of inducing hypertension in nonhypertensive mice. Myeloid Vdr deletion promotes vascular macrophage infiltration, increases ROS-mediated scavenging of NO, and reduces renal perfusion, resulting in increased plasma renin. Interestingly, deletion of myeloid Vdr also promotes macrophage infiltration into the JG cell apparatus, thereby regulating JG cell renin production by an unexpected communication between macrophages and JG cells through miR-106b-5p. Moreover, knockout of miR-106b-5p prevents the elevation in BP and stimulation of JG cell renin production by KODMAC macrophages, suggesting myeloid-specific effects. In diet-induced vitamin D deficiency, miR-106b deletion prevents vitamin D-deficiency-induced, renin-dependent hypertension in mice, and macrophages from such mice are unable to activate JG cell renin production or release. These findings provide evidence of an intercellular miRNA communication mechanism enabling macrophages of the innate immune system to induce renin production in JG cells and promote hypertension.

Recently, miRNAs have emerged as regulators of vascular inflammation and RAS-induced hypertension[25]. Interestingly, in two different rat hypertensive models, plasma miR-106b expression robustly increased during the development of hypertension and concomitant heart failure[34]. In this study, we provide confirmation via multiple mouse models that vitamin D-deficient macrophages secrete increased miR-106b-5p, which enters into JG cells and downregulates the transcription-related factors Pde3b and E2f1, leading to increased renin production and release. A previous study has shown that the PDE3 inhibitor trequinsin increases cellular cAMP content and stimulates JG cell renin release[35]. The miR-106b-5p-mediated downregulation of PDE3B and the associated increase in adenylate cyclases, cAMP, and CREB abundance suggest that cAMP signaling is an important contributor to miR-106b-5p-induced renin production[36]. Moreover, deletion of E2f1[26] or Pde3b[27] in mice promotes increased Ppargc1a, Ppars, and Esrra gene expression in tissues requiring a high level of energy for normal physiological function; all these transcription factors are known to increase renin release in JG cells[37,38]. These data suggest that miR-106b in JG cells is a global modulator linking cell signaling pathways to transcription factors implicated in renin production and release, and may provide an important target to ameliorate renin-dependent hypertension.

Macrophages display a spectrum of phenotypes including "M1" and "M2" in response to different stimuli, and miRNA expression profiles may reflect these macrophage phenotypes[39,40].

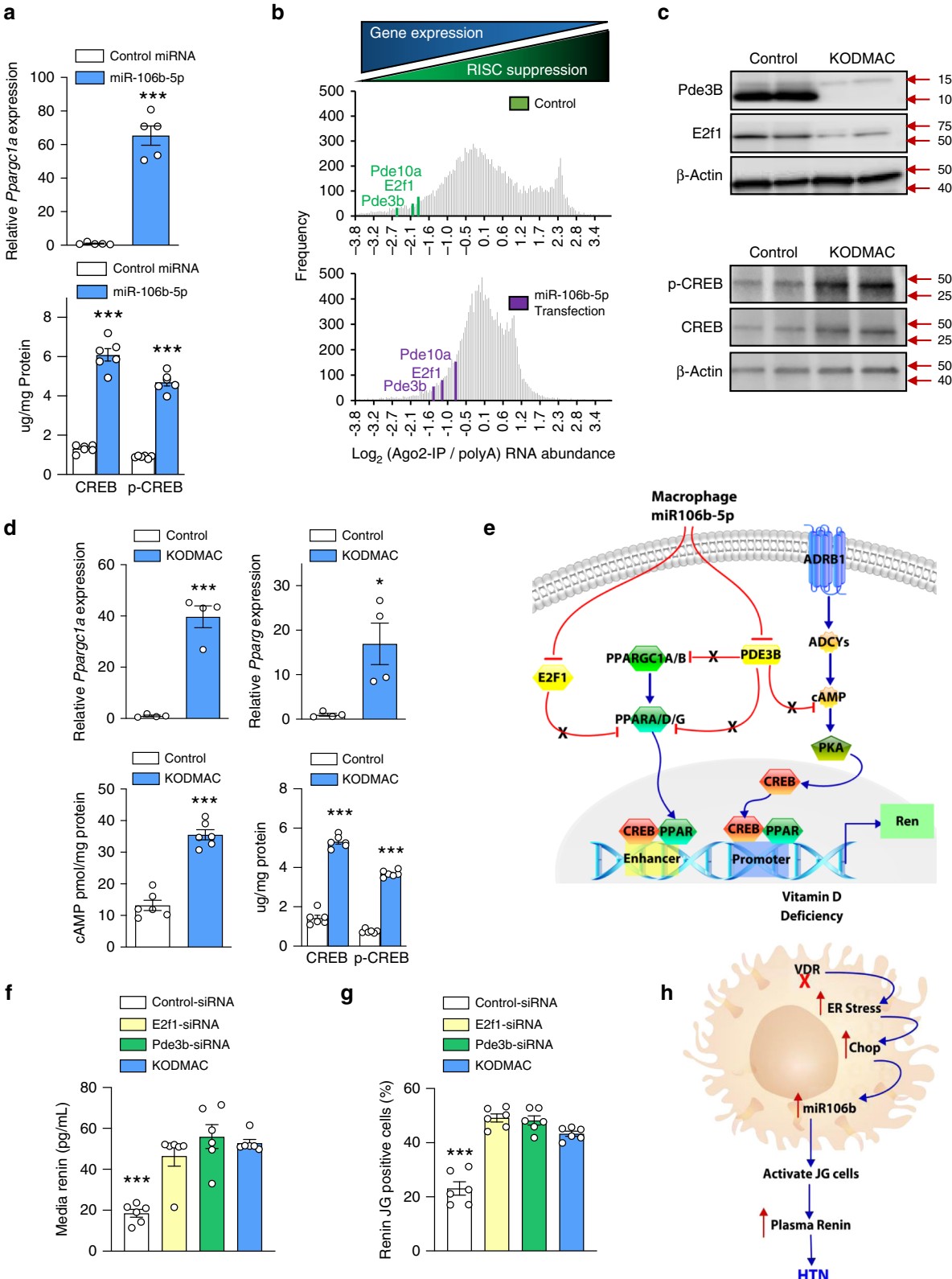

However, the mechanism(s) that regulates altered macrophage miRNA secretion are unknown. We previously demonstrated that regulation of ER stress by vitamin D in type 2 diabetes exerts a pivotal role in monocyte recruitment, macrophage phenotype, and cholesterol metabolism, and regulates critical components of the vascular remodeling seen in diabetic vascular disease[13,21,22]. A recent study indicated that ER stress activation in HeLa cells increased multivesicular body formation and exosome release by the ER stress transducers IRE1 and PERK, suggesting that ER stress not only regulates exosome release, but might also change miRNA secretion profiles[24]. In this study, we found that activation of macrophage PERK-unfolded protein response by *Vdr* deletion promotes miR-106-5p exosome release and causes JG cell renin production, independently of vitamin D status. Moreover,

**Fig. 4 Enhancement of renin transcription by miR-106b-5p modulation of PPAR and CREB signaling. a** Pppargc1a mRNA expression and CREB ($n = 5$/group) and phospho-CREB protein abundance in miR-106b- (light-blue bar), or control- (white bar) transfected JG cells ($n = 6$/group). **b** Histogram displaying frequencies of $\log_2$ Ago2-IP vs. global polyA(RNA) abundance ratio for all detected mRNAs (gray bars) with positions of selected PPAR and CREB signaling mediators under control (green) and miR-106b-transfected (purple) conditions. **c** E2f1, Pde3b, CREB, and phospho-CREB protein expression in JG cells exposed to control (white bar) or KODMAC media (light-blue bar) (representative of 4/group). **d** Pppargc1a and Pparg mRNA expression ($n = 4$/group), and cAMP and CREB protein abundance in JG cells after culture with KODMAC or control peritoneal macrophage media ($n = 6$/group). **e** Schematic diagram of relevant cell signaling pathways. Macrophage miR-106b-5p inhibits JG cell E2f1 and Pde3b, removing inhibition of PCG1 and CREB pathways to induce renin production. Blue arrows represent stimulatory pathways and red lines represent inhibitory pathways, while X represents repressed inhibition. **f** Renin-positive (YFP-labeled Ren[1c]) JG cells and **g** renin secretion after transfection with siRNAs against E2f1 (yellow bar) or Pde3b (green bar) or coculture with KODMAC macrophage media (light-blue bar) ($n = 6$/group). **h** Mechanistic schematic diagram. Data expressed as mean ± SEM from (**a**, **d**) Student's two-tailed unpaired $t$ test with \*$P < 0.05$, \*\*\*$P < 0.001$ vs. control and (**f**, **g**) one-way ANOVA with Tukey's post hoc test with \*\*$P < 0.001$ vs. all.

global deletion of the ER stress chaperone Ddit3 (CHOP) prevents vitamin D-deficiency-induced renin-dependent hypertension in mice, and their macrophages are unable to activate JG cell renin production. These data indicate a critical role of the macrophage pPERK/CHOP-unfolded protein response in regulating inflammation-associated metabolic diseases.

Our findings provide evidence that dysregulated macrophage signaling in response to vitamin D deficiency is sufficient to cause hypertension by a microRNA-specific mechanism that enables communication from innate immune cells to JG cells. A detailed understanding of the mechanisms by which macrophage ER stress causes miR-106b-5p secretion will direct the development of specific therapies preventing the miRNA-mediated activation of multiple signaling pathways and transcription factors involved in renin-dependent hypertension.

## Methods

**Experimental models**. Mice with inactivation of VDR in myeloid cells (KnockOut of vitamin D receptor in MACrophages: KODMAC) were generated and used for assessment of blood pressure and macrophage–JG cell interactions. Briefly, $Vdr^{fl/fl}$ mice were crossed with mice expressing transgenic Cre recombinase under control of the Lyzosomal M promoter ($Lyz2Cre^{+/-}$)(Jax/Lab 004781) in the $Ldlr^{-/-}$ (Jax/Lab 002207) background as well as in C57BL6/J (Jax/Lab 000664) and $Apoe^{-/-}$ (Jax/Lab 002052) backgrounds[13]. Littermate mice with $Lyz2Cre^{-/-}Vdr^{fl/fl}$ in the corresponding background were used as controls. To evaluate the role of macrophage miR-106b secretion and/or ER stress on vitamin D-deficiency-induced hypertension, we also studied miR-106b$^{-/-}$ (Jax/Lab 008460) and Ddit3$^{-/-}$ (Jax/Lab 030816) mice compared to C57BL6/J littermates. Vitamin D-sufficient and -deficient models were generated by feeding standard chow (Harlan TD7022) or vitamin D-deficient chow diet with 2% calcium (Harlan TD87095), respectively, for 8 weeks prior to assessment. 25-hydroxyvitamin D levels in the dietary vitamin D-deficient models were as follows (deficient vs. sufficient, respectively): 9.4 ± 1.7 vs. 51 ± 2.8 in C57, 48 ± 4 vs. 47 ± 2.1 in KODMAC, 7.6 ± 0.7 vs. 44 ± 6 in miR-106b$^{-/-}$, and 13 ± 1 vs. 51 ± 2.3 in Ddit3$^{-/-}$. To generate the KODMAC/miR-106$^{-/-}$ double knockout, we first crossed $Vdr^{fl/fl}Cre^{+/-}$ with miR-106b$^{-/-}$ mice until we obtained $Vdr^{fl/fl}Cre^{+/-}$/miR-106$^{-/-}$, $Vdr^{fl/fl}Cre^{+/-}$/miR-106b$^{+/+}$, or control $Vdr^{+/+}$/miR-106b$^{+/+}$ mice, all in the C57BL6 45.2 background. Specific $Vdr$ deletion was verified[13]. Then, we transplanted their BM into C57BL6 45.1 recipients and characterized their phenotype 8 weeks post transplant. Mice were housed in a clean facility with an ambient temperature of 65–75 °F, 40–60% humidity, and 12 light/12 dark cycles. All experiments included male and female animals. Protocols followed were approved by the Washington University Institutional Animal Care and Use Committee and compliant with ethical regulations for studies involving laboratory animals.

**Bone marrow (BM) transplantation**. BM cells were isolated from the femurs and tibias of 8–10-week-old mice by flushing the bones with cold PBS[13]. The total bone marrow was washed, triturated using a 24-gauge needle (Benson Dickson), collected by centrifugation at $200 \times g$ for 4 min, and diluted with PBS. After lysis of erythrocytes using 0.05% sodium azide, cells were counted to obtain a defined concentration of unfractionated bone marrow. Eight-week-old recipient mice were lethally irradiated with 10 Gy from a cesium-137 γ-cell irradiator. Within 6 h after irradiation, recipients were reconstituted with ~$5 \times 10^6$ donor marrow cells via a single injection. The donor marrow was allowed to repopulate for 8 weeks after transplantation prior to assessment. For estimating engraftment, the total bone marrow was isolated from the femurs and tibias from C57BL/6-TgN (GFP-positive) (Jax/Lab 003291) mice in the same background in parallel for each transplant experiment. FACS quantification of the percentage of GFP positivity in peripheral

leukocytes revealed engraftment of 85–92%. In BM recipients of KODMAC/miR-106b$^{-/-}$ or KODMAC/miR-106b$^{+/+}$, the degree of engraftment was performed by using C57BL6 45.2 donors and 45.1 recipients to confirm engraftment in the experimental animals (CD45.1 and CD45.2 antibodies, Biolegend, Inc). Flow-cytometry quantification of the percentage of 45.2 positivity in peripheral leukocytes or recipients revealed engraftment of 88–96%, implying reduction in $Vdr$ or miR-106b expression to 4–12% of baseline.

**Vascular physiological assessment**. Systolic and diastolic blood-pressure measurements were measured in conscious mice 8 weeks after weaning using the CODA-2 noninvasive tail cuff measurement system. Mice of both genders were acclimated to handling and placement in the apparatus daily for 2 days before the measurement of blood pressure. Multiple measurements were made at each of three daily sessions and averaged to obtain a single value for each mouse[41]. Twenty-four-hour invasive blood-pressure monitoring in conscious mice was performed using the PA-C10 telemetry sensor by the Data Science International System. Mouse BP recording was completed a week after the sensor implantation. Blood-pressure data obtained were averaged every 30 min for 24 h, including the mouse light and dark cycles[42]. Vascular $O_2\cdot^-$ formation was measured by the lucigenin-enhanced chemiluminescence assay. Mouse thoracic aortas were removed, placed into prewarmed Krebs–Henseleit/HEPES buffer, and cut into 3-mm ring segments. Segments were transferred to a white 96-well microplate containing Krebs–Henseleit/HEPES buffer (100 μL) with lucigenin (10 μmol/L; this low concentration prevents autooxidation). NADPH (100 μmol/L) was added to each well after 10 min. Chemiluminescence was recorded every 2 min in a microplate luminometer at 37 °C (Lucy-1, Rosys Anthos). The vessels were dried (24 h at 90 °C) and weighed. The results are expressed as counts per minute per milligram dry tissue[43]. Vascular responses to inhibition of nitric oxide synthase were recorded invasively (PowerLab/8SP, AD Instruments) after the intravenous infusion of 5 mg kg$^{-1}$ L-NAME[42]. Renal blood flow was performed under isoflurane anesthesia, using a renal flow probe (0.5PSB Nanoprobe with handle, Transonic) in 8-week-old mice[44] after volume repletion with 8 μl/g body weight of normal saline, followed by 0.5 μl/min/g.

**Immunofluorescence**. Kidneys and aortas were harvested, and 10-μm cryosections from distal aorta and kidney were fixed for 24 h in 10% formalin and blocked with goat antiserum. Then, they were incubated with primary MOMA-2 rat monoclonal antibody dilution 5 μg/ml (Abcam #ab33451), and/or anti-renin[45] overnight, then with secondary antibody. DAPI (Vector Laboratories) mounting was used for counterstaining. Images were captured on a confocal microscope (Olympus Fv1000)[29]. Vascular and renal macrophages were quantified by flow cytometry using two different macrophage markers (CD11b and F4/80)[46,47]. To perform tissue flow cytometry, we extracted mouse aortas (heart to the iliac bifurcation) and decapsulated kidneys and minced into 2–4-mm pieces using scissors or scalpel blade. Both tissues were mixed with collagenase media on a rotary tube suspension mixer at 37 °C for 20 min and then mechanically digested using a 1000-μL pipette tip. Disperse cells were filtered through a cell strainer and collected in 10 mL of Flow Cytometry Staining Buffer. A single-cell suspension was generated by pressing with the plunger of a 3-mL syringe. Cells were then centrifuged at $300–400 \times g$ for 4–5 min at 2–8 °C. The cell pellet was resuspended in flow cytometry staining buffer. A final cell concentration of $1 \times 10^7$ cells/mL was incubated with 0.2 mg/mL of C11b-PE (eBioscience™ #12-0112-81) and F4/80-PECy7 (eBioscience™ 25-4801-82) antibodies for 15 min on ice, then washed, and flow cytometry was performed. Cell aggregates, dead, and cellular debris were excluded based on FSC/SCC. Batch analysis by FlowJo version 9.6.2 was used for gating consistency, and select positive population. We used unstained samples and blocking with FC to decrease autofluorescence and unspecific background. Cells were stained with two macrophage antibodies for anti-CD11b PE and F4/80 PE-Cyanine7 F4/80. Rat IgG2b K (eBioscience™ #12-4031-82) and IgG2a K (eBioscience™ #25-4321-82) isotype control for anti-CD11b PE and F4/80 PE-Cyanine7, respectively, to ensure antibody-specific detection. The oxidative fluorescent probe dihydroethidium (DHE) and confocal microscopy were used to detect

in situ levels of ROS in aorta sections. Ten-micron unfixed frozen sections were used for DHE visualization with adjacent sections analyzed by DAPI to detect all mural nuclei. To confirm that DHE fluorescence reflects ROS levels, sections from aortae were preincubated for 30 min with superoxide dismutase–polyethylene glycol (10 mmol/L) before incubation with DHE. Images were digitally captured by fluorescence microscopy as described[48]. In the kidney, MOMA-positive macrophages were quantified by the percentage of the total renal area. Images were acquired by an observer blinded to the treatment groups.

**Peritoneal macrophages and juxtaglomerular (JG) cells**. Murine peritoneal macrophages were collected immediately following injection of PBS into the peritoneum[13,29]. Macrophages were cultured in DMEM plus 10% FBS for 3 h prior to assessment. Ren1$^c$-YFP JG cells were cultured in DMEM/F12 media with 10% exosome-depleted FBS[17]. Cells were seeded at $0.1 \times 10^6$ cells/well in 12-well plates and grown to 80–90% confluence, estimated to be a maximum of $0.5 \times 10^6$ cells/well. Transwell chambers were utilized (Costar polycarbonate filters, 3-μm pore size). Membranes and 12-well plates were coated with fibronectin (5 μg/mL, Life Technologies)[13] overnight at 4 °C[13]. Macrophages ($0.3 \times 10^5$ cells/well) were added to the upper chamber, or directly over the Ren1$^c$-YFP JG cells cultivated in the lower chamber. Cells were incubated for 72 h, and quantification of renin production was performed using computer-based quantification using Image J to identify YFP-positive JG cells across different samples. We binarized our images to decrease low-intensity objects, and used an erode filter to remove objects smaller than a cell. We generated black-and-white images for DAPI and YFP channels, counted the cells that were positive for both channels, quantified the number of cells positive for DAPI, or both YFP and DAPI, and expressed as a percentage of all DAPI-positive cells. Macrophage media and JG cells were additionally incubated with 0.2 μg/ml TNFα-neutralizing antibody on 1 μg/mL dilution (R&D Biosystems #MAB4101) for 72 h, and quantification of renin-producing cells was evaluated as described above. Reduction of ER stress was obtained by adding a chemical chaperone, phenylbutyric acid (PBA, 20 mmol/L, Calbiochem, San Diego, CA) to peritoneal vitamin D-deficient macrophages for 18 h prior to coculture. Induction of ER stress was obtained by adding thapsigargin (0.25 nmol/L, Sigma) to peritoneal macrophages from vitamin D-sufficient mice for 18 h prior to coculture. YFP-labeled Ren1$^c$-JG cells were cocultured for 72 h with media from PBA-treated vitamin D-deficient or thapsigargin-treated vitamin D-sufficient macrophages[21].

**Metabolic assays**. Renin was measured in plasma samples pooled from groups of five animals by mouse renin-1 ELISA kit from Sigma-Aldrich (#RAB0565). TNF-α (BD Biosciences) and IL-1β (R&D Biosystems) concentrations were measured by ELISA in coculture media[13]. Urine Na sample was obtained by a bladder catheter after volume repletion with 8 μl/g body weight of normal saline, followed by 0.5 μl/min/g and analyzed by electrolyte analyzer (Roche). Total and phosphorylated CREB abundance was measured by magnetic bead-based immunoassay (Bio-Rad Bio-Plex #171V60013M and #171304006 M). cAMP level was measured by ELISA (Invitrogen #EMSCAMPL).

**MicroRNA array**. RNA was extracted from cell culture supernatants (macrophage media), processed, and hybridized to Affymetrix GeneChip miRNA 4.0 microarrays by Washington University's Genome Technology Access Center. Post-processing of array signal data was performed with Partek Genomics Suite 6.6 (Partek, St Louis, MO).

**Exosome isolation**. We isolated exosome vesicles from media from macrophages alone and from macrophage coculture with YFP-labeled Ren1$^c$-JG cells using ultracentrifugation. Following 72 h of coculture, media was collected, then centrifuged at $2000 \times g$ for 30 min to remove cells and debris. The supernatant was then transferred to a new tube without disturbing the pellet. The media was then centrifuged at $100,000 \times g$ for 18 h. Density-gradient-based isolation was then performed to obtain purer exosome preparations using the Total Exosome Isolation kit (Invitrogen)[49,50]. The degree of purity of the isolated exosome vesicles was confirmed by western blot of 0.5 μg of protein per lane showing the presence of EV marker protein expression from categories 1a and 1b [Invitrogen CD63 (#PA5-92370), CD81 (#MA5-32333), and CD9 (#MA5-31980)] and category 2a [Invitrogen Alix (#PA5-52873)] and the absence of non-EV protein expression from category 3a [Invitrogen albumin (#PA5-89332)][50,51].

**MicroRNA and mRNA expression via RT-qPCR**. miRNA purification and isolation from Ren1$^c$-YFP JG cells was performed using the *mir*Vana miRNA kit (Invitrogen Ambion) and anti-miR (Life Technologies #AM10067). RTq-PCR was performed using the TaqMan reagent kit, with relative expression of miRNA calculated by the comparative threshold cycle method relative to miR-U6 in cell tissue as an internal control. For assays from culture media, we spiked in miR-39 from *C. elegans* as an exogenous housekeeping miRNA control prior to extraction (Qiagen #219610). Relative expression of miRNA was calculated by the comparative threshold cycle method relative to miR-39 abundance. TaqMan primers were obtained from Life Technologies for miRNAs 17-5p (#002308), 17-3p (#002543), 106b-5p (#000442), 106b-3p (#002380), 26a-5p (#000405), 26a-3p (#002443), 146a-5p (#000468), 146a-3p (#463191), 195a-5p (#000494), 195a-3p (#002107),

103-5p (#121218), 103 3p (#000439), 99b-5p (#000436), 99b-3p (#002196), 107-5p (#465082), 107-3p (#000443), 155-5p (#002571), 155-3p (#464539), 151-5p (#002642), 151-3p (#001190), 340-5p (#002258), 340-3p (#002259), 191-5p (#002299), 191-5p (#002576), 130b-5p (#002460), 130b-3p (#00456), 92a-5p (#002496), 92a-3p (#00430), U6 (#4427975,) and 39 (#467942). For mRNA RTq-PCR, total RNA (1 μg) was treated with DNase and reverse-transcribed using Superscript II (Invitrogen) with oligo-dT as the primer. PCR was performed with the GeneAmp 7000 Sequence Detection System using the SYBR® Green reagent kit (Applied Biosystems). RNA not subjected to reverse transcription was included in each assay as negative control. We used the following mouse oligonucleotides: for *Pparg* forward 5′-AGCCTCATGAAGAGCCTTCCA-3′, *Pparg* reverse 5′-TCCGGAAGAAACCCTTGCA-3′; *Pparg*c1 forward 5′-AGCCTCTTTGCCCA-GATCTT-3′, *Pparg*c1 reverse 5′-GGCAATCCGTCTTCATCCAC-3′; *Ren1* forward 5′-TTACGTTGTGAACTGTAGCCA-3′, *Ren1* reverse 5′-AGTATGCA-CAGGTCATCGTTC-3′; *Pde3b* forward 5′-GGTGATGGTGGTGAAGAA-3′, *Pde3b* reverse 5′-AGTGAGGTGGTGCATTAG-3′; *E2F1* forward 5′-ACTCCTCGCAGATCGTCATCATCT-3′, *E2f1* reverse 5′-GGACGTTGGT-GATGTCATAGATGCG-3′; *Mrpl32* forward, 5′-AAGCGAAACTGGCGGAAAC-3′, *Mrpl32* reverse, 5′-GATCTGGCCCTTGAACCTTCT-3′. All assays were done in triplicate. Data are expressed as relative expression of mRNA normalized to mouse ribosomal protein *Mrpl32*.

**Plasmids and small-interfering RNA transfection**. Ren1$^c$-YFP JG cells were transfected with *mir*VR-106b mimic (Life Technologies #4464066) and miR-106b antagomir (Life Technologies #AM10067), and 72 h after transfection, mRNA expression in JG cell lysates was evaluated with RT-qPCR, and JG cell renin production was evaluated with confocal microscopy (Olympus Fv1000). Ren1$^c$-YFP JG cells were transduced with lentivirus containing sense in *Pde3b*-siRNA, *E2f1*-siRNA, or control siRNA for 48 h (Life Technologies #s71385, #65234, and #4390843, respectively)[20]. Protein expression for PDE3B (Abcam AB #95814, dilution 1 μg/mL), E2F1 (Abcam #112580, dilution 1 μg/mL), CREB-1 (Santa Cruz #SC377154, dilution 5 μg/mL), and phospho-CREB-1 (Ser133, Cell Signalling #9198, dilution 1 μg/mL) was performed in JG cells exposed to 72 h of KODMAC or control media. Renin mRNA expression and renin production were determined 72 h after recovery from transfection or viral transduction. Also, Ren1$^c$-YFP JG cells were transfected with pri-miR-106b-5p siRNA sense oligonucleotides 5′-CCU AAU GAC CCU CAA GCC GUU-3′ and antisense 5′-CGG CUU GAG GGU CAU UAG GUU-3′, then exposed to KODMAC macrophage media for 72 h, and miR-106 pre-miR or mature miR-106b mRNA was assessed.

**RNA-seq and RISC-seq methods**. Transcriptomic profiling of individual JG cell samples was performed using RNA sequencing beginning with 1 mg of total RNA isolated using Trizol (Life Technologies) exactly according to the manufacturer's directions as initial input for mRNA-Seq, or with Ago2-immunoprecipitated material from 150-mm-diameter dishes for RISC-Seq. For mRNA-seq, polyA RNA enrichment was performed using Dynabeads mRNA Purification Kit (Life Technologies, Carlsbad, CA, USA). Ago2 immunoprecipitation was performed in ice-cold 50 mM Tris–HCl, 5 mM EDTA, 5 mM EGTA, pH 7.5, with 0.5% Nonidet P-40, Roche Complete protease inhibitors, and yeast tRNA and SUPERnase-IN added to final concentrations of 1 μg/μL and 1 U/μL[52]. polyA-RNA and Ago2-immunoprecipitated RNA was fragmented using a fragmentation buffer (40 mM Tris acetate, 100 mM K acetate, and 30 mM Mg acetate, pH 8.2; 2 min for 30 s at 95 °C). First-strand cDNA was generated using the Superscript III 1$^{st}$ Strand Synthesis System (Life Technologies) and the second strand was generated using 10 U DNA Polymerase I (New England Biolabs, Ipswich, MA, USA), 40 U *E. coli* DNA Ligase (New England Biolabs), and 2 U RNase H (Life Technologies) in 20 mM Tris.HCl, pH 6.9, 66 μM dNTPs, 90 mM KCl, 4.6 mM MgCl$_2$, 0.15 mM b-NAD$^+$, and 10 mM (NH$_4$)$_2$SO$_4$ at 16 °C for 2 h. Finally, Illumina sequencing libraries were prepared using End-It DNA End Repair Kit (Epicentre, Madison, WI, USA) and Klenow Fragment (30/50 exo-) (New England Biolabs) for the ligation of Illumina sequencing adapters (Illumina-Index-AdapterA and 5′-phosphorylated Illumina-Index-AdapterB) using the Liga-Fast™ Rapid DNA Ligation System (Promega, Madison, WI, USA). Fragments between 150 and 300 bp were recovered after 2% agarose gel electrophoresis in order to remove adapter dimers by size selection (QIAquick Gel Extraction Kit, Qiagen). Nucleotide indexes to permit sample deconvolution were added to individual libraries using Phusion DNA polymerase (New England Biolabs) during 16 cycles of amplification. Samples were quantitated (Nanodrop, Thermo Scientific, Wilmington, DE, USA), mixed, and submitted for sequencing on a HiSeq 3000 instrument (Illumina Inc., San Diego, CA, USA) at Washington University GTAC to obtain single-end, 50-bp reads. After removal of mouse ribosomal RNA reads using a custom procedure, the remaining reads were aligned to the Illumina iGenomes GRCm38 Ensembl release of the mouse transcriptome using Tophat2 version 2.1.1[53]. An average of $5.2 \times 10^6$ aligned reads were obtained per sequencing library and quantitated at the gene level using HTSeq version 0.9.1[54]. mRNAs were included in downstream analyses if present in at least three of four biological replicates in any experimental group (polyA control, polyA miR-106b, Ago2-IP control, and Ago2-IP miR-106b) at an abundance of at least 1 read per million. A total of 15,365 mRNAs were obtained from this filtering procedure. Pairwise comparisons between experimental groups were made using the robust dispersion modification of edgeR version 1.2.4[55].

**Statitical analysis**. Experiments were carried out in duplicate or triplicate. The number of samples in each experiment (*n*) refers to the number of distinct samples. Gaussian distribution of continuous variables was verified by Kolmogorov–Smirnov distance. Parametric data are expressed as the mean ± SEM and analyzed by two-sided *t* tests, paired or unpaired as appropriate, or by one-way ANOVA and Tukey's post test for more than two groups. Nonparametric data are presented as the median and analyzed using the Mann–Whitney test. Differences were considered statistically significant if $P \le 0.05$. Statistical analysis was carried out using GraphPad Prism version 8.4.3.

**Reporting summary**. Further information on research design is available in the Nature Research Reporting Summary linked to this article.

## Data availability

Noncoding RNA-profiling array data, as well as RNA sequences from JG cells and RISC sequence data have been deposited in the NCBI GEO repository with accession numbers GSE 155511 and GSE117704, respectively. Source data are provided with this paper.

## Code availability

There is no custom computer code or algorithm used to generate the results in this paper. Source data are provided with this paper.

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

## Acknowledgements

This work was supported by NIH R01HL09481806, VA Merit Award 1BX003648-01. The contents of this article are solely the responsibility of the authors and do not necessarily represent the official view of NIH.

## Author contributions

J.O., S.J.M., A.E.R., G.C., M.S.S., R.A.G., R.A.S.-L., and C.B.-M. planned the experiments. J.O., S.M.B., and J.S.S. performed animals' noninvasive blood-pressure experiments. A.B. and B.A.B., microRNA isolation and immunohistochemistry. M.S.S. performed trans-plant experiments. R.H.K. and R.P.M. performed the L-Name experiments and analyzed the results. P.O.-O. and K.J.B. performed and analyzed renal ultrasound data. H.Z. and C. G.N. performed and analyzed the invasive blood-pressure data. R.D.H. and R.A.B. performed and analyzed the gene array experiments. S.J.M. performed the RISC experiments and analysis. Finally, S.J.M., A.E.R., M.S.S., R.A.G., R.A.S.-L., and. C.BM. wrote and revised the paper. All authors have given approval to the final version of the paper. Figure 4e was generated with ePath3D (ProteinLounge Inc) by R.A.B. and Fig. 4h was created by C.B.-M.

## Competing interests

The authors declare no competing interests.
