## [Peer Review File · Nature Communications]

Reviewer #1 (Remarks to the Author):

Ms. NatComm

Title: Macrophage Secretion of miR-106b-5p Causes Renin-Dependent Hypertension

General Comments

That vitamin D deficiency is linked to hypertension has been reported in many epidemiological and observational studies, but most clinical trials using vitamin D to control blood pressure or cardiovascular diseases have failed. In animal studies, vitamin D deficiency or VDR deficiency leads to renin-dependent high blood pressure, as vitamin D acts to suppress renin gene expression, but this mechanism has not been confirmed in humans. So there are a lot of controversies surrounding this topic. In this context, this manuscript provides evidence that vitamin D deficiency or VDR deficiency in macrophages induces hypertension through the renin-angiotensin system in mice. The data show that VDR-depleted macrophages are inflammatory and infiltrates into the kidney to induce renin release from the JG cells. The mechanism is that vitamin D deficiency or VDR deletion induces ER stress to release miR-106b through exosome, which is taken up by JG cells. miR-106b somehow stimulates renin expression. This work includes in vitro and in vivo evidence to support the conclusion, but there are many serious issues that need to be clarified.

Specific Comments

1. One critically important data is blood pressure. Non-invasive BP measurement usually has a lot of problems. Invasive 24 hr telemetric BP monitoring is considered as a gold standard. Throughout the paper the only invasive BP data is Figure S1A, but this data only shows the average BP, which is not acceptable. At least a 24 hr continuous BP tracing in these two genotypes of mice should be shown.
2. The baseline renin activity is so different among all the assays. For example, in Figure 1, the baseline is 20 in 1B, but becomes 10 even 5 in 1J. So in the BM transplantation, the difference between the genotypes is because of much lower baseline renin in the control, not really because of real increase caused by KOD BM cells. Why the baselines are so different? Indeed, in any animals, a 4x renin activity increase is unimaginable (Fig. 1J). It is questionable that macrophages, not considered as a major physiological renin regulator, can increase renin that much.
3. Supplemental Figure 2A. Is it a renal glomerulus? Both renin and CD68 staining appear just background. The staining for renin and macrophages appears too much to be true. Some 2nd antibody control should be shown.
4. Renal blood flow is decreased as shown in Figure 1H. If it's true, that can trigger renin release and increase systemic renin. How much systemic renin and BP changes are contributed from this?
5. It is claimed that JG cells uptake miR-106b secreted from macrophages via exosomes. There is no evidence of that. It's better to suppress exosome transport. This is a major weakness of this work. If that's the case, what happens to the endogenous miR-106b in the JG cells? Can macrophages secrete a factor to stimulate miR-106b in JG cells? There is no data to demonstrate miR-106b silence in Fig. 2K and L.
6. Figure 4 is confusing and another major weakness. miRNAs regulate (usually decrease) protein and RNA levels for their target transcripts, but in Figure 4A miR-106b induces a large portion of the mRNA list. By what mechanism? The miR-106b-suppressed mRNAs include CREB. It is well established that cAMP-PKA-CREB pathway is a major signal to maintain and up-regulate renin expression, but CREB mRNA is reduced here in Figure 4A, which does not fit into the renin increase theory in mice with VDR-deleted macrophages. Data in Figure 4B suggest Esf1, Pde3b mRNAs are under less RISC suppression by miR-106b? It's unclear. The scheme in Figure 4C is unclear either; it does not seem to explain why renin is up-regulated. Overall how miR-106b induces renin is not

well explained.

7. Page 5 line 20. "significantly reduce" or "significantly induce"?
8. Page 8 line 6. "KODMAC macrophages" seem to be "VD deficient macrophages" in Figure 3E.
9. "HPF" should be defines in Figure 2.
10. Evidence for nutritional vitamin D deficiency should be presented in all the VD deficient animals.

Reviewer #2 (Remarks to the Author):

This study investigates the role of the vitamin D receptor (VDR) in macrophages on renin-dependent hypertension. Compared to controls, LDLR^{-/-} mice with myeloid cell-specific deletion of VDR (KODMAC) have (1) elevated BP and PRAs, (2) increased vascular infiltration of macrophages, DHE stains, and ROS detection, (3) blunted renal cortical blood flow and hypertensive responses to L-NAME. Transplant of WT bone marrow lowers renin levels and BP. KODMAC macrophages or media increase renin fluorescence in JG YFP reporter cells. Among miRNAs isolated from KODMAC macrophage media, transfection of JG cells with miRNA 106b-5p most prominently enhances renin positivity in JG cells. KODMAC macrophage media can increase levels of this miRNA in JG cells that have been silenced for endogenous 106b-5p. mir-106b deficiency prevents the hypertension and renin induction induced by a VD-deficient diet, and transplant of mir-106b^{-/-} bone marrow reduces BP and renin in VD-deficient WT mice. Blocking ER stress in KODMAC macrophages reduces 106b production. Ddit^{-/-} mice and bone marrow chimeras are protected from ER stress and hypertension induced by VD deficiency. Mir-106b-5p represses E2f1 and Pde3b protein in JG cells. Knockdown of mRNA for these proteins in JG cells increases renin production. These studies are novel and creative and elucidate an axis through which macrophages and JG cells may interact to drive BP elevation. The experiments are comprehensive, but following are opportunities to strengthen the manuscript.

1. The authors should verify specific deletion of VDR in the myeloid cells of their conditional mutants by qPCR and acknowledge that this strategy will hit other myeloid populations including neutrophils.
2. The urinary sodiums should be measured after a change in diet. Otherwise the mice should be in sodium balance. Similarly, renal perfusion is typically measured in response to a vasoconstrictor.
3. The authors should quantitate aortic macrophages by flow to support the contention that there is increased vascular macrophage infiltration in the KODMAC cohort. Similarly, they should quantitate renal macrophage infiltration since the macrophage – JG cell interaction underpins their hypertension paradigm.
4. For the bone marrow transplant experiments, the authors should demonstrate the efficacy of the bone marrow transfer by measuring gene expression for VDR or mir-106b as appropriate in splenocytes of the bone marrow recipients.

Reviewer #3 (Remarks to the Author):

Macrophage secretion of miR106b-5p causes renin-dependent hypertension
Oh J, Matkovich SJ, ...Bernal-Mizrachi C

In this paper by Oh et al. they followed up on a previous study in which they showed that loss of vitamin D receptor in macrophages induces ER stress, and leads to increased cholesterol uptake and atherosclerosis. In the present study the authors investigated the potential link between loss of VitD and hypertension, induced by macrophages modulating JG cells in the kidney.

While the story is quite interesting and extends on previous observations that macrophages play a crucial role in regulating the RAS system and blood pressure, the main weakness of the paper

stems from the fact that the authors compare VDR fl/fl mice (macrophage specific knockout) with full KO for several other mediators (Ddit3, miR106b), and then make quite strong claims about the role of these signaling pathways in macrophages specifically, which is at least misleading. Furthermore, based on the data presented it is absolutely unclear whether these miRs are indeed transferred via exosomes as suggested by the authors. Therefore more carefully controlled experiments are needed to validate the strong claims the authors are making throughout the paper. In its present form, the paper is therefore not suitable for publication in Nat Comm.

Major comments:

Figure 2 Panels 2F and 2G

-I lack any information on these presumed exosomes. How were they isolated, characterized? I could not find any information in the material and methods section, which was rather remarkable. I would like to refer to Van Deun J, et al. EV-TRACK: transparent reporting and centralizing knowledge in extracellular vesicle research. Nature methods. 2017; 14(3):228-32. which insists on characterizing EVs thoroughly as to get a more transparent communication on what people define throughout literature as "exosomes", and to clarify the field. Similarly, last year the MISEV Guidelines have been proposed with a similar goal J Extracell Vesicles. 2018 Nov 23;7(1):1535750. doi: 10.1080/20013078.2018.1535750. eCollection 2018

- Panels 2F and 2G I was quite struck by the massive differences in expression of miR between these two panels and trying to understand how we could compare both figure panels? How do the authors compare the amount of material present in both systems? How is normalization done?
- Panels 2F and 2G : In this respect, there does not seem to be any correlation between the amount of miRs in exosomes (massive expression of miR195a-5p for example) and whether it is detectable in media. Any explanation for this?
- Panels 2F and 2G : miR106b-5p appears to be hardly induced in media derived from VitD suff versus VitD def macrophages in comparison with all the others, suggesting it could only minorly contribute to exert effects on JG cells. I have a hard time to understand how such a minor (2-fold) induction would mediate such strong effects in all the subsequent experiments and why lack of this specific miR would be sufficient to abolish all effects of conditioned media on JG cells (Fig.3B).
- I am also quite puzzled by the massive difference in percentage of renin positive cells in steady state control mice throughout the different figures. This seems to range from 40% (Fig. 3B Vit D Suff -Vit D suff, which should be control mice), to +/-28% (Fig. 3H) to 5% in Fig. 2H. How is this scored and where is the huge variation between individual JG cells coming from? How do the authors interpret small differences in renin positive JG cells (like in Fig. 3B) with these huge variations in control conditions in background. In general, is there a way to score renin positivity in a more quantitative way? For me it was also not clear what these cells were, a JG cell line, primary cells transduced with Ren-YFP?
- How do the authors know that the miR 106 levels measured in Fig 2J are coming from the JG cells, not macrophages, since they were cocultured as far as I understood.
- Panel I: did the authors take a similar approach for other miRs as well (eg miR26a-5p; miR193a-3p...) to show that this antagonistic effect was specific for miR106b? Because, if this is not the case, then how explain the strong effect of the miR106b-/-?
- Panel J, the fold induction between the unstimulated JG cells versus JG cells cocultured with macrophages was striking as it seems larger than the fold induction caused by the stimulus that would lead to miR106 production. How is this explained? Is there constitutive miR106bp presence? What is the relative abundance of all these miRs in steady state macrophages? due to the normalization of the steady state to 1 in all panels showing miR expression, this is hard to judge.

-I think overall it is really not clear to me how we can be sure that these miR are indeed derived from macrophage exosomes. Is it possible to do simple experiments like adding conditioned media with and without exosomes to show that the effect is exosome dependent? Is it possible to interfere with exosome secretion by using certain (specific) drugs that interfere with secretion? Or alternatively, could one use dicer-/- macrophages to prevent miR formation in macrophages and use those as a source for JG stimulation? The dicerKO mice should be crossed onto KODMAC mice,

see also further. If one would use the dicer KO mice, one could compare the effect of dicer with the miR10b KO macrophages as to determine the relative importance of miR106b.

-Figure 3

-my main concern for this whole figure is that the authors are comparing systemic processes versus macrophage specific processes and then deduce macrophage specific functions, which is quite confusing. both the mir106b and the ChopKO are full body KO and then used in BM adoptive experiments in mice that were either Vit D deficient or sufficient, based on their diet. How does irradiation affect the whole process of vitamin D deficiency and hypertension? Why did the authors not cross the ChopKO or miR106b KO onto the VDRfl/fl LysM Cre mice, and then from those mice use BM for adoptive transfer in WT mice. With this system one is at least sure that the defects are in one and the same cell type and one can make more claims about certain pathways being able to restore defects in other pathways, eg loss of chop or miR 106 compensates for the loss of Vdr in macrophages.

- Is Vit D deficiency specifically inducing activation of PERK? the authors should show which ER stress signaling pathways are activated in macrophages, both upon genetic deficiency of Vdr as upon diet induced loss of VitD. Are these pathways also triggered in the JG cells (in VitD deficient diet conditions)?

-PBA should be validated to show to confirm that lower levels of miR106 are indeed caused by lowering levels of PERK activation and Chop induction

-How to explain that Chop leads to induction of miR106b-5? A previous paper by the group of Afshin Samali showed downregulation of miR106b-25 cluster by PERK. What is the difference?

Specific Comments

1. One critically important data is blood pressure. Non-invasive BP measurement usually has a lot of problems. Invasive 24 hr telemetric BP monitoring is considered as a gold standard. Throughout the paper the only invasive BP data is Figure S1A, but this data only shows the average BP, which is not acceptable. At least a 24 hr continuous BP tracing in these two genotypes of mice should be shown.

Answer: We agree with the reviewer and have now included 24h BP tracings across 3 animals per group in Figure 1A, demonstrating the consistent BP difference between KODMAC and control mice.

2. The baseline renin activity is so different among all the assays. For example, in Figure 1, the baseline is 20 in 1B, but becomes 10 even 5 in 1J. So in the BM transplantation, the difference between the genotypes is because of much lower baseline renin in the control, not really because of real increase caused by KOD BM cells. Why the baselines are so different? Indeed, in any animals, a 4x renin activity increase is unimaginable (Fig. 1J). It is questionable that macrophages, not considered as a major physiological renin regulator, can increase renin that much.

Answer: We agree with the reviewer. Because our study has been conducted over several years, the renin assays performed were not always consistent. To address this, we repeated all of the measurements using a single assay and performed all the measurements at the same time from same saved samples. The baseline results are now more consistent, and the differences described previously are still significant. We modified the figures accordingly.

3. Supplemental Figure 2A. Is it a renal glomerulus? Both renin and CD68 staining appear just background. The staining for renin and macrophages appears too much to be true. Some 2nd antibody control should be shown.

Answer: We agree with the reviewer that this figure was somewhat confusing, so we eliminated it as it was confirmation of Figure 2B.

4. Renal blood flow is decreased as shown in Figure 1H. If it's true, that can trigger renin release and increase systemic renin. How much systemic renin and BP changes are contributed from this?

Answer: It is difficult to isolate the specific contribution of decreased renal blood flow in the hypertensive phenotype in these mice. We postulate that KODMAC macrophage infiltration into the vasculature and the kidney mediates both reduced renal blood flow and miR-106b-mediated JG cell production of renin, making it difficult to isolate these respective contributions as they both derive from the same immune cells. However, we did prove that in culture conditions, KODMAC macrophages secrete miR-106b to induce JG cell renin production, suggesting at least some contribution of this mechanism to the phenotype.

5. It is claimed that JG cells uptake miR-106b secreted from macrophages via exosomes. There is no evidence of that. It's better to suppress exosome transport. This is a major weakness of this work. If that's the case, what happens to the endogenous miR-106b in the JG cells? Can macrophages secrete a factor to stimulate miR-106b in JG cells? There is

no data to demonstrate miR-106b silence in Fig. 2K and L.

Answer: We appreciate the reviewer's comment. To prove that the KODMAC macrophages were the source of the JG cell miR106b after stimulation, we pre-treated the JG cells by transfecting with siRNA for pre-miR-106b so these cells cannot produce mature miR-106b-5p (we clarified the figure legends for updated Figures 2K and S4C to specify that we used the siRNA). We found that the pre-miR106b-siRNA-transfected JG cells exposed to KODMAC media had similarly high levels of mature miR-106b-5p compared to control-siRNA-transfected cells exposed to KODMAC media without increasing pre-miR-106b-5p expression, suggesting that mature miR-106b was transferred from KODMAC media and not generated from JG cells (supplemental Fig. 4C). To confirm the specificity of miR-106b-5p from KODMAC macrophages as the driver of JG cell renin production, we generated myeloid cells with double knockout of Vdr and miR-106b. Then, JG cells were exposed to media from peritoneal macrophages of KODMAC miR-106b^{-/-}, KODMAC miR106b^{+/+}, or WT mice. We found that the absence of miR-106b in KODMAC macrophages suppressed induction of JG cell renin production compared to KODMAC cells with intact miR-106b-5p, confirming the specificity of KODMAC miR-106b-5p for this phenotype (updated Figures 3E-3G).

6. Figure 4 is confusing and another major weakness. miRNAs regulate (usually decrease) protein and RNA levels for their target transcripts, but in Figure 4A miR-106b induces a large portion of the mRNA list. By what mechanism? The miR-106b-suppressed mRNAs include CREB. It is well established that cAMP-PKA-CREB pathway is a major signal to maintain and up-regulate renin expression, but CREB mRNA is reduced here in Figure 4A, which does not fit into the renin increase theory in mice with VDR-deleted macrophages. Data in Figure 4B suggest Esf1, Pde3b mRNAs are under less RISC suppression by miR-106b? It's unclear. The scheme in Figure 4C is unclear either; it does not seem to explain why renin is up-regulated. Overall how miR-106b induces renin is not well explained.

Answer: We agree with the reviewer's comment. We have now simplified Figure 4 to make it clearer. While we acknowledge that some of the CREB isoforms were downregulated by miR-106b in the RNA seq results (former Figure 4A and S6), we are now reporting total CREB abundance (including all the isoforms), which is elevated (updated Figure 4D). We additionally now show that miR-106b-5p increased cAMP levels in JG-YFP cells. We measured cAMP in JG-YFP cells exposed to macrophage media from KODMAC or control mice. We found a significant increase in cAMP levels in JG-YFP cells exposed to macrophage KODMAC media compared to control media (updated Figure 4D). These findings, along with increased expression of adenylyl cyclase 3, 4, 5, 6, 7, 9 (enzymes generating cAMP and important in stimulation of renin secretion, Figure S6), and decreased expression of Pde3b and Pde10 (cAMP degradation enzymes, updated Figure 4C) support that miR106b activation of cAMP-PKA is a potential pathway stimulating JG cell renin production. In Figure 4B, E2f1, Pde3b, and Pde10a have increased suppression (i.e. lower expression) in response to miR-106b-5p. We have clarified the figure to convey this more clearly. We also simplified the scheme in now Figure 4E to specify the effects of miR-106b-5p on the various pathways.

7. Page 5 line 20. "significantly reduce" or "significantly induce"?

Answer: We agree with the reviewer and changed the sentence to “significantly induce”.

8. Page 8 line 6. “KODMAC macrophages” seem to be “VD deficient macrophages” in Figure 3E.

Answer: We agree with the reviewer and changed the sentence.

9. “HPF” should be defines in Figure 2.

Answer: We included the definition in the legend for Figure 2.

10. Evidence for nutritional vitamin D deficiency should be presented in all the VD deficient animals.

Answer: We added the vitamin D levels to the methodology.

Rev 2

1. The authors should verify specific deletion of VDR in the myeloid cells of their conditional mutants by qPCR and acknowledge that this strategy will hit other myeloid populations including neutrophils.

Answer: KODMAC Vdr expression was published in Cell Rep. 2015 Mar 24;10(11):1872-86 and these experiments were performed concurrently with those reported in this publication. We modified the methodology to reflect this. We changed the language in a number of paragraphs throughout the manuscript to address that the lack of Vdr is not specific to monocyte/macrophages but to myeloid cells.

2. The urinary sodium should be measured after a change in diet. Otherwise the mice should be in sodium balance. Similarly, renal perfusion is typically measured in response to a vasoconstrictor.

Answer: We neglected to mention that mice were volume replete with normal saline infusion prior to assessment of urine electrolytes and renal blood flow. The methodology has been edited to reflect this. Additionally, before we evaluated renal blood flow, we measured the change in invasive systolic or diastolic BP in response to different doses of intravenous angiotensin II and phenylephrine using a PowerLab/8SP instrument, but we did not find any differences between the groups. Therefore, we did not assess renal perfusion in response to vasoconstriction.

3. The authors should quantitate aortic macrophages by flow to support the contention that there is increased vascular macrophage infiltration in the KODMAC cohort. Similarly, they should quantitate renal macrophage infiltration since the macrophage – JG cell interaction underpins their hypertension paradigm.

Answer: As requested by the reviewer, vascular and renal macrophages were quantified by flow cytometry using two different macrophage markers (CD11b and F4/80). This data is now included in Figures S1F and S2A.

4. For the bone marrow transplant experiments, the authors should demonstrate the efficacy of the bone marrow transfer by measuring gene expression for VDR or mir-106b as appropriate in splenocytes of the bone marrow recipients.

Answer: For our original experiments, we verified engraftment by performing parallel transplants from GFP-positive C57 donors into recipients identical to each experimental condition that were GFP-negative. Additionally, for some of these models, we verified engraftment by using C57BL6 45.2 donors and 45.1 recipients. For all additional transplants for this revision (KODMAC/miR106b^{-/-}), we utilized 45.2 donors and 45.1 recipients to verify engraftment. Flow cytometry quantification of the percentage of 45.2 positivity in peripheral leukocytes or recipients revealed engraftment of 88%–96%, thus indirectly implying reduction in Vdr or miR106b expression to 4-12% of baseline. We clarified the methodology regarding verification of BM engraftment in the methodology section.

Rev 3

In this paper by Oh et al. they followed up on a previous study in which they showed that loss of vitamin D receptor in macrophages induces ER stress, and leads to increased cholesterol uptake and atherosclerosis. In the present study the authors investigated the potential link between loss of VitD and hypertension, induced by macrophages modulating JG cells in the kidney.

While the story is quite interesting and extends on previous observations that macrophages play a crucial role in regulating the RAS system and blood pressure, the main weakness of the paper stems from the fact that the authors compare VDR fl/fl mice (macrophage specific knockout) with full KO for several other mediators (Ddit3, miR106b), and then make quite strong claims about the role of these signaling pathways in macrophages specifically, which is at least misleading. Furthermore, based on the data presented it is absolutely unclear whether these miRs are indeed transferred via exosomes as suggested by the authors. Therefore more carefully controlled experiments are needed to validate the strong claims the authors are making throughout the paper. In its present form, the paper is therefore not suitable for publication in Nat Comm.

Major comments:

1. Figure 2 Panels 2F and 2G
-I lack any information on these presumed exosomes. How were they isolated, characterized? I could not find any information in the material and methods section, which was rather remarkable. I would like to refer to Van Deun J, et al. EV-TRACK: transparent reporting and centralizing knowledge in extracellular vesicle research. Nature methods. 2017;14(3):228-32. which insists on characterizing EVs thoroughly as to get a more transparent communication on what people define throughout literature as “exosomes”, and to clarify the field. Similarly, last year the MISEV Guidelines have been proposed with a similar goal J Extracell Vesicles. 2018 Nov 23;7(1):1535750. doi: 10.1080/20013078.2018.1535750. eCollection 2018

Answer: We appreciate the reviewer comments. Following the reviewer’s suggestion, we isolated

exosomes from media from macrophages and from macrophage co-culture with YFP-JG cells using the protocol in the recommended paper by ultracentrifugation followed by density-gradient-based isolation and repeated all miRNA qPCR analysis from the extracted exosomes., The degree of purity of the isolated exosome vesicles was confirmed by showing the presence of EV marker proteins from category 1a and 1b (CD63, CD81 and CD9) and category 2a (Alix) and the absence of non-EV protein expression from category 3a (albumin) in our samples as described by the MISEV guidelines (Supp. Fig. 3B). We updated the methodology section with further details.

2. Panels 2F and 2G I was quite struck by the massive differences in expression of miR between these two panels and trying to understand how we could compare both figure panels? How do the authors compare the amount of material present in both systems? How is normalization done?
 - Panels 2F and 2G :In this respect, there does not seem to be any correlation between the amount of miRs in exosomes (massive expression of miR195a-5p for example) and whether it is detectable in media. Any explanation for this?
 - Panels 2F and 2G :miR106b-5p appears to be hardly induced in media derived from VitD suff versus VitD def macrophages in comparison with all the others, suggesting it could only minorly contribute to exert effects on JG cells. I have a hard time to understand how such a minor (2-fold) induction would mediate such strong effects in all the subsequent experiments and why lack of this specific miR would be sufficient to abolish all effects of conditioned media on JG cells (Fig.3B).

*Answer: Incorporating this result with the reviewer's prior comment, we wondered whether the variability in media exosome miRNA expression seen in Figures 2F and 2G was due to the technique of exosome isolation. Therefore, we repeated the experiments and isolated exosomes by ultracentrifugation followed by density-gradient-based isolation as recommended above from media of KODMAC, control, vitamin D-deficient, and vitamin D-sufficient macrophages and then repeated miRNA expression analyses by qPCR of miRNAs identified by the unbiased array. The normalization for qPCR analysis of the media exosomes was performed by spiking a known amount of miR-39 from *C. elegans* as an exogenous housekeeping miRNA control. Relative expression of miRNA was calculated by the comparative threshold cycle method relative to miR-39 abundance. We found that not only was the variability of miRNA expression significantly reduced, but miR-106b-5p was the most abundant miRNA from both vitamin D deficient and KODMAC macrophage media (~30-40 fold increase vs. controls). There were still differences in the quantifications of some miRNAs between the KODMAC and vitamin D deficient exosomes compared to their relevant controls, but this is likely related to the differences in the systems models. It is possible that in KODMAC mice lacking of VDR during embryogenesis promoted different adaptive compensatory mechanism compared to postnatally vitamin D deficiency. Because dietary vitamin D-deficiency is more reflective of a clinical condition, we chose to pursue miR-106b-5p because of its consistent upregulation across both systems. We updated Figures 2F and 2G with the new data.*

3. I am also quite puzzled by the massive difference in percentage of renin positive cells in steady state control mice throughout the different figures. This seems to range from 40% (Fig 3B Vit D Suff -Vit D suff, which should be control mice), to +/-28% (Fig. 3H) to 5% in Fig. 2H. How is this scored and where is the huge variation between individual JG cells coming from? How do the authors interpret small differences in renin positive JG

cells (like in Fig. 3B) with these huge variations in control conditions in background. In general, is there a way to score renin positivity in a more quantitative way? For me it was also not clear what these cells were, a JG cell line, primary cells transduced with Ren-YFP?

Answer: We agree with the reviewer comments. In order to standardize quantification of renin production, we re-analyzed all our co-culture experiments using computer-based quantification by binarizing our images using Image J to identify YFP-positive JG cells across different samples (include description in the method section). This approach gave us a more consistent assessment of the fluorescence quantification. We modified all figures that include renin quantification. Regarding the origins of the Ren-YFP-JG cells, these cells were isolated from mice in which the coding sequence for YFP was cloned into the translation initiation site of the Ren1^c promoter. They were generated by our collaborators and co-authors on this manuscript, Ariel Gomez and Maria Luisa Sequeira-Lopez, and have been used in multiple publications (Am J Physiol Heart Circ Physiol. 2008 Feb;294(2):H699-707; J Clin Invest. 2018;128(11):4787-4803).

4. How do the authors know that the miR 106 levels measured in Fig 2J are coming from the JG cells, not macrophages, since they were cocultured as far as I understood?

Answer: A similar concern was raised by reviewer 1, and we addressed this in reviewer 1, comment 5 above. Please also refer to comment 1 for reviewer 3 regarding exosome isolation.

5. Panel I: did the authors take a similar approach for other miRs as well (eg miR26a-5p;miR193a-3p...) to show that this antagonistic effect was specific for miR106b? Because, if this is not the case, then how explain the strong effect of the miR106-b/-?

Answer: We initially focused on miR-106b-5p because of its differential levels in KODMAC macrophage media compared to control, combined with its association in the literature with hypertension. We have not fully tested the other differentially expressed miRNAs, and it is very possible that some of them also have an effect on blood pressure. However, by itself, miR-106b-5p has a strong enough effect to result in physiologically detectable changes, as evidenced by the decreased blood pressure and lack of macrophage stimulation of JG cell renin production in transplant recipients of KODMAC/miR106b^{-/-} bone marrow donors (updated figures 3E-3G).

6. Panel J, the fold induction between the unstimulated JG cells versus JG cells cocultured with macrophages was striking as it seems larger than the fold induction caused by the stimulus that would lead to miR106 production. How is this explained? Is there constitutive miR106bp presence? What is the relative abundance of all these miRs in steady state macrophages? due to the normalization of the steady state to 1 in all panels showing miR expression, this is hard to judge.

Answer: We did not test all of the miRs in unstimulated JG cells, but did quantify miR-106b-5p and noted very little baseline abundance. While there was an increase in miR-106b-5p abundance after exposure to control macrophages (or media), the increase after exposure to KODMAC macrophages (or media) was greater. Given the intercellular signaling role of macrophages, we were not surprised by some increase in miR-106b-5p with control macrophage exposure, but it was the KODMAC macrophage exposure that resulted in the physiologic blood pressure response. We did quantitate the abundance of all of the other miRs in JG cells after

exposure to KODMAC vs. control macrophage media. The non-normalized results are below and demonstrate similarly low levels of all of the miRs analyzed after control macrophage exposure, but the greatest increase with KODMAC macrophage exposure was in miR-106b-5p.

miR	Control Media Δ CCT	Control Media $2^{(-\Delta$ CCT) $*10000$	KODMAC Δ CCT	KODMAC Media $2^{(-\Delta$ CCT) $*10000$	Fold differences
miR 106-5p	12.53	2.09	7.94	43.38	20.76
miR 155-5p	13.19	1.40	8.84	23.03	16.46
miR193a-5p	12.04	2.82	8.15	43.01	15.27
miR99-5p	11.98	3.61	8.17	37.73	10.46
miR17-3p	13.59	0.87	9.86	12.70	14.57
miR103-5p	10.99	5.78	7.38	64.50	11.16
miR340-5p	13.31	1.42	9.65	13.30	9.37
miR107-5p	11.20	8.41	7.54	54.30	6.45
miR325-5p	10.82	5.68	8.24	34.36	6.05

7. I think overall it is really not clear to me how we can be sure that these miR are indeed derived from macrophage exosomes. Is it possible to do simple experiments like adding conditioned media with and without exosomes to show that the effect is exosome dependent? Is it possible to interfere with exosome secretion by using certain (specific) drugs that interfere with secretion? Or alternatively, could one use dicer-/- macrophages to prevent miR formation in macrophages and use those as a source for JG stimulation? The dicer KO mice should be crossed onto KODMAC mice, see also further. If one would use the dicer KO mice, one could compare the effect of dicer with the miR10b KO macrophages as to determine the relative importance of miR106b.

Answer: We appreciate the reviewer's suggestions. In Figure 2J, we demonstrate that exposure of JG cells to exosome-depleted media compared to macrophages or their media results in much lower miR106b-5p abundance, suggesting it is resulting from macrophage exposure. We confirmed using the exosome isolation methodology suggested by this reviewer that miR-106b-5p was higher in KODMAC macrophage exosomes. Based on new KODMAC/miR106b^{-/-} macrophage results (see reviewer 1, response 5), we know that the JG cell renin production is miR106b-dependent. Finally, the data of JG cell miR-106b silencing with pre-miR-106b siRNA indicates that the JG cell renin production was secondary to exogenous macrophage miR-106b-5p. It is difficult to design the suggested experiments to further confirm whether these KODMAC effects are exosome-dependent. Drugs that interfere with exosome secretion from macrophages seem to impair inflammatory cytokines (Biochim Biophys Acta. 2015;1852:2362-71). Similarly, conditional deletion of dicer in myeloid cells modifies the inflammatory properties, promoting alternative macrophage activation, macrophage apoptosis and accumulation of lipids when fed high fat diet (Circulation. 2018;138:2007-2020). These alterations could independently introduce changes in the KODMAC macrophage phenotype, making it difficult to determine whether the phenotype is an exosome-specific effect using these models.

8. Figure 3. my main concern for this whole figure is that the authors are comparing systemic processes versus macrophage specific processes and then deduce macrophage specific functions, which is quite confusing. both the mir106b and the ChopKO are full body KO and then used in BM adoptive experiments in mice that were either Vit D deficient or sufficient, based on their diet. How does irradiation affect the whole process of vitamin D deficiency and hypertension? Why did the authors not cross the ChopKO or miR106b KO onto the VDRfl/fl LysM Cre mice, and then from those mice use BM for adoptive transfer in WT mice. With this system one is at least sure that the defects are in one and the same cell type and one can make more claims about certain pathways being able to restore defects in other pathways, eg loss of chop or miR 106 compensates for the loss of Vdr in macrophages.

Answer: We would like to thank the reviewer for this insightful comment. It is known that irradiation prior to BMT in humans is associated with an increased risk of vitamin D deficiency long-term, but it not known whether this is causal. Similarly, irradiation and BMT in mice is associated with increased angiotensin II-induced vascular inflammation, and BMT nephropathy in humans can cause hypertension. However, all of our transplant experiments included control recipients who underwent identical irradiation prior to transplantation, so we expect that the effects of radiation are accounted for in our study design and do not detract from our findings. Per this reviewer's suggestion, we have now generated double knockout (KODMAC/miR-106b^{-/-}) mice for comparison to KODMAC/miR-106^{+/+} and Vdr^{+/+}/miR-106b^{+/+} mice and transplanted their BM into WT mice as suggested. We found that knockout of miR-106b prevented the elevation in BP and stimulation of JG-YFP renin production by KODMAC macrophages, suggesting myeloid-specific effects, which was the primary goal of these experiments (Figure 3E-G).

9. Is Vit D deficiency specifically inducing activation of PERK? the authors should show which ER stress signaling pathways are activated in macrophages, both upon genetic deficiency of Vdr as upon diet induced loss of VitD. Are these pathways also triggered in the JG cells (in VitD deficient diet conditions)?

Answer: In our previous publications (Oh J et al. Cell Rep. 2015 Mar 24;10(11):1872-86; Riek AE et al. J Biol Chem. 2012;287(46):38482-94; and Oh J et al. J Biol Chem. 2012;287(15):11629-4), we found that lack of Vdr in macrophages as well as in macrophages exposed to vitamin D deficient conditions have increased both PERK and IRE phosphorylation, as well as CHOP protein expression by reducing SERCA function. This is an intriguing question, as there is a dearth of publications evaluating the effect of ER stress activation on renin production by JG cells. We also found no evidence in the literature that vitamin D deficiency activates ER stress in JG cells. We did not specifically explore in our JG cells whether vitamin D deficiency induced renin production via PERK or IRE1. However, in our CHOP knockout experiments, vitamin D deficient knockout macrophages prevented JG renin production in transwell co-cultures, while vitamin D deficient macrophages with intact CHOP induced JG renin production (Figure 3M). If vitamin D deficiency was inducing ER stress in JG cells to activate renin production, we would expect increased renin even in the setting of co-culture with CHOP knockout vitamin D deficient macrophages. However, we did not assess in vivo whether vitamin D deficiency induced JG cell renin production by activation of ER stress.

10. PBA should be validated to show to confirm that lower levels of miR106 are indeed caused by lowering levels of PERK activation and Chop induction

Answer: We appreciate this comment, and we have validated the efficacy of PBA suppressing PERK α and Chop expression in vitamin D-deficient macrophages in our previous publications (Riek AE et al. J Biol Chem. 2012; 287(46):38482-94 and Oh J et al. J Biol Chem. 2012;287(15):11629-4). These references are included in the manuscript.

11. How to explain that Chop leads to induction of miR106b-5? A previous paper by the group of Afshin Samali showed downregulation of miR106b-25 cluster by PERK. What is the difference?

Answer: In this study the authors observed a significant down regulation of the miR-106b-25 cluster in three non-immune cell lines after a prolonged ER stress stimulation with two ER stress inducing agents (thapsigargin or tunicamycin). ER stress and the unfolded protein responses (UPR) are unique from cell to cell. Additionally, in Dr. Samali's study, the authors did not evaluate whether changes in the intracellular miR-106b levels are due to increased miR-106b secretion via exosomes, thus depleting intracellular levels. Previous studies have indicated that exosome release is increased during ER stress via IRE1- and PERK-mediated pathways (Kanemoto, S Biochem Biophys Res Commun. 2016 Nov 11;480(2):166-172). Therefore, it is possible that the results are not contradictory, but that they are in different cell types and assessing different locations of the miRNA in response to ER stress.

Reviewer #1 (Remarks to the Author):

Ms. Nat Communications

Title: Macrophage Secretion of miR-106b-5p Causes Renin-Dependent Hypertension

This revision has addressed many of the issues raised in the review, but the question of how miR-106 regulates renin remains confusing. The authors agreed that CREB is a major renin promoter and the authors claimed that miR-106 increases CREB signaling leading to renin increase. But the data in Figure S6 clearly show that miR-106 transfection suppressed CREB1/3/5 mRNAs, so it is unclear why the total CREB protein was increased in Figure 4A (no explanation or discussion). The measurement method of CREB protein was not clearly described. The information about the antibody should be provided (only state immunoassays), and really Western blot data for CREB should be presented to strengthen the claim (as in Fig. 4C). Also Figure 4E is confusing. One could not understand what it means. The legend is too simple to understand. I assume it is the result of miR-106 transfection, but it needs to be clearly explained. Why the two proteins E2f1 and PDE3b inhibit the other proteins? Again it does not explain why CREB protein is up-regulated.

Reviewer #2 (Remarks to the Author):

The authors have addressed this reviewer's concerns.

Reviewer #3 (Remarks to the Author):

The authors have responded to the previous reviews with extensive new experimental data and the manuscript presents novel and convincing data.

Dear Editor,

We are grateful that two of the reviewers felt that their concerns had been appropriately addressed for publication. The manuscript has been revised according to the suggestions and comments of reviewer 1. The major revised parts are highlighted in red color for your convenience of re-reviewing.

The responses to the specific comments of the reviewer 1 are as follows:

Reviewer 1 : His revision has addressed many of the issues raised in the review, but the question of how miR-106 regulates renin remains confusing. The authors agreed that CREB is a major renin promoter and the authors claimed that miR-106 increases CREB signaling leading to renin increase. But the data in Figure S6 clearly show that miR-106 transfection suppressed CREB1/3/5 mRNAs, so it is unclear why the total CREB protein was increased in Figure 4A (no explanation or discussion). The measurement method of CREB protein was not clearly described. The information about the antibody should be provided (only state immunoassays), and really Western blot data for CREB should be presented to strengthen the claim (as in Fig. 4C). Also Figure 4E is confusing. One could not understand what it means. The legend is too simple to understand. I assume it is the result of miR-106 transfection, but it needs to be clearly explained. Why the two proteins E2f1 and PDE3b inhibit the other proteins? Again it does not explain why CREB protein is up-regulated.

Answer: We acknowledge the reviewer's comments. Figure S6 is a result of RNA sequence data, so not necessarily indicative of protein expression (1-3). When we assayed protein, we found that both total and phosphorylated CREB were elevated using a Bio-Plex antibody (Bio-Rad Bio-Plex #171V60013M and # 171304006M). However, we also performed a Western blot for CREB1 (Santa Cruz antibody #SC-377154, detects between amino acids 254-327 near the C-terminus) and for phospho-CREB1 (Cell Signaling antibody #9198 that detects serine 133) and confirmed that both were elevated in JG cells in the setting of miR-106b-5p stimulation. We believe that this data, in conjunction with that showing elevated downstream adenylate cyclase expression and cAMP abundance proves activation of JG cell CREB pathways by miR-106b-5p. We modified the results section accordingly and described with more detail in the methodology. We modified Figure 4E to emphasize that macrophage miR-106b-5p drives JG cell pathways that ultimately induce renin production. It has been previously shown that E2F1 and PDE3B inhibit PGC1/PPAR, cAMP, and CREB pathways, and this inhibition is removed in the setting of miR-106b-5p exposure.

1. W. H. Walker, C. Girardet, J. F. Habener, Alternative exon splicing controls a translational switch from activator to repressor isoforms of transcription factor CREB during spermatogenesis. *J Biol Chem* **271**, 20145-21050 (1996).
2. Y. Liu, A. Beyer, R. Aebersold, On the Dependency of Cellular Protein Levels on mRNA Abundance. *Cell* **165**, 535-550 (2016).
3. S. T. Crews, J. C. Pearson, Transcriptional autoregulation in development. *Current biology : CB* **19**, R241-246 (2009).

REVIEWERS' COMMENTS:

Reviewer #1 (Remarks to the Author):

Major concerns addressed. No more comments.